# Addressing caveats of neural persistence with deep graph persistence

**Leander Girrbach***  
*University of Tübingen, Tübingen AI Center*  
*leander.girrbach@uni-tuebingen.de*

**Anders Christensen***  
*Technical University of Denmark*  
*University of Tübingen, Tübingen AI Center*  
*andchri@dtu.dk*

**Ole Winther**  
*Technical University of Denmark*  
*University of Copenhagen*  
*Copenhagen University Hospital*  
*FindZebra*  
*olwi@dtu.dk*

**Zeynep Akata**  
*University of Tübingen, Tübingen AI Center*  
*zeynep.akata@uni-tuebingen.de*

**A. Sophia Koepke**  
*University of Tübingen, Tübingen AI Center*  
*a-sophia.koepke@uni-tuebingen.de*

**Reviewed on OpenReview:** *https://openreview.net/forum?id=oyfRWeoUJY*

## Abstract

Neural Persistence is a prominent measure for quantifying neural network complexity, proposed in the emerging field of topological data analysis in deep learning. In this work, however, we find both theoretically and empirically that the variance of network weights and spatial concentration of large weights are the main factors that impact neural persistence. Whilst this captures useful information for linear classifiers, we find that no relevant spatial structure is present in later layers of deep neural networks, making neural persistence roughly equivalent to the variance of weights. Additionally, the proposed averaging procedure across layers for deep neural networks does not consider interaction between layers. Based on our analysis, we propose an extension of the filtration underlying neural persistence to the whole neural network instead of single layers, which is equivalent to calculating neural persistence on one particular matrix. This yields our deep graph persistence measure, which implicitly incorporates persistent paths through the network and alleviates variance-related issues through standardisation. Code is available at https://github.com/ExplainableML/Deep-Graph-Persistence.

## 1 Introduction

Analysing deep neural networks to gain a better understanding of their inner workings is crucial, given their now ubiquitous use and practical success for a wide variety of applications. However, this is a notoriously difficult problem. For instance, neural networks often generalise well, although they are overparameterised (Zhang et al., 2017). This observation clashes with intuitions from classical learning theory. Both the training process, which is a stochastic process influenced by minibatches and modifications of gradient-based weight updates due to the chosen optimiser, e.g. Adam (Kingma & Ba, 2015), and the computations performed by

---

*Denotes equal contribution.

a trained neural network are very complex and not easily accessible to theoretical analysis (Goodfellow & Vinyals, 2015; Choromanska et al., 2015; Hoffer et al., 2017).

*Topological Data Analysis (TDA)* has recently gained popularity for analysing machine learning models, and in particular deep learning models. TDA investigates data in terms of its scale-invariant topological properties, which are robust to perturbations (Cohen-Steiner et al., 2007). This is a desirable property in the presence of noise. In particular, *Persistent Homology (PH)* provides means of separating essential topological structures from noise in data. We refer to (Pun et al., 2022) for an overview of the field, as a detailed introduction to TDA and PH is out of the scope of this work.

Recent works consider neural networks as weighted graphs, allowing analysis with tools from TDA developed for such data structures (Rieck, 2023). This is possible by considering the intermediate feature activations as vertices, and parameters as edges. The corresponding network weights are then interpreted as edge weights. Using this perspective, Rieck et al. (2019) define *neural persistence*, a popular measure of neural network complexity, which is calculated on the weights of trained neural networks, i.e. the edges of the computational graph. Neural persistence does not consider the data and intermediate or output activations. Nevertheless, Rieck et al. show that neural persistence can be employed as an early stopping criterion in place of a validation set. We wish to comprehend which aspects of neural networks are measured by inspection tools such as neural persistence. This helps understanding the limitations and developing new applications for the methods used for model assessment. In addition, unresolved problems that require further theoretical justification can be exposed. In this paper, we attempt to conduct such an analysis for neural persistence.

Our results suggest that neural persistence is closely linked to the variance of weights and is better suited to analyse linear classifiers than deep feed-forward neural networks (DNNs). As the possibility of applying TDA for analysing DNNs is highly desirable, we propose to modify the complexity measure and introduce *deep graph persistence* (DGP) which extends neural persistence to also capture inter-layer dependencies. The proposed extension alleviates certain undesirable properties of neural persistence, and makes DGP applicable in the downstream task of covariate shift detection. This task has been tackled by an earlier extension of neural persistence, called *topological uncertainty* (Lacombe et al., 2021). Our results indicate our proposed extension to be better suited for detecting image corruptions.

To summarise, our contributions are as follows: (1) We identify the variance of weights and spatial concentration of large weights as important factors that impact neural persistence. (2) We observe that later layers in deep feed-forward networks do not exhibit relevant spatial structure and demonstrate that this effectively reduces neural persistence to a surrogate measure of the variance of weights. (3) We introduce *deep graph persistence*, a modification of neural persistence, which takes the full network into account and reduces dependency on weight variance, making it better suited for deep networks.

## 2 Related work

Topological data analysis has recently gained popularity in the context of machine learning (Carrière et al., 2019; Hofer et al., 2017; 2019; Hu et al., 2019; Khrulkov & Oseledets, 2018; Kwitt et al., 2015; Moor et al., 2020b; Ramamurthy et al., 2019; Rieck et al., 2019; Rieck, 2023; Royer et al., 2020; Zhao & Wang, 2019; Reininghaus et al., 2015; Stolz et al., 2017; Rieck, 2023; von Rohrscheidt & Rieck, 2022; Birdal et al., 2021; Hajij et al., 2023). For a recent, in-depth overview of the field we refer to (Zia et al., 2023). Topological features have been used as inputs to machine learning frameworks, or to constrain the training of models. For instance, topological features were extracted from leukaemia data to predict the risk of relapse Chulián et al. (2023) and to improve the classification of fibrosis in myeloproliferative neoplasms Ryou et al. (2023), and persistence landscapes were incorporated into topological layers in image classification (Kim et al., 2020). Additionally, McGuire et al. (2023) studied the learnt internal representations for topological feature input. Different to this, Moor et al. (2020b;a) constrain the training to preserve topological structures of the input in the low-dimensional representations of an autoencoder, and Chen et al. (2019) introduce a penalty on the topological complexity of the learnt classifier itself. Recently, He et al. (2023) proposed a topology-aware loss for instance segmentation which uses persistence barcodes for the predicted segmentation and for the ground truth, aiming at maximising the persistence of topological features that are consistent with the ground truth.

Another prominent line of work uses TDA to analyse neural networks, either by considering the hidden activations in trained neural networks, or by examining the network weights. Corneanu et al. (2020); Gebhart et al. (2019); Ballester et al. (2022) interpret hidden units as points in a point cloud and employ TDA methods to analyse them. For example, Corneanu et al. (2020) use the correlation of activations for different inputs as a distance metric and observe linear correspondence between persistent homology metrics and generalisation performance. Naitzat et al. (2020) find that learning success is related to the simplification of the topological structure of the input data in the hidden representations of trained networks. In contrast, Wheeler et al. (2021) propose to quantify topological properties of learnt representations. To facilitate model inspection or understand finetuning of word embeddings, Rathore et al. (2021; 2023) introduce visualisation tools for learnt representations based on TDA. Chauhan & Kaul (2022) formulate a topological measure that links test accuracy to the hidden representations in language models. Different to those works, we analyse the structure in the weights of trained neural networks.

Several complexity measures for the weights of DNNs have been proposed, e.g. by considering the weights as weighted stratified graphs (Rieck et al., 2019), or analysing the learnt filter weights in convolutional neural networks (Gabrielsson & Carlsson, 2019). Birdal et al. (2021) also consider network weights and analyse the network's intrinsic dimension, linking it to generalisation error. Generalisation has also been shown to be directly linked to a model's loss landscape (Horoi et al., 2022). Rieck et al. (2019) introduce *neural persistence*, a complexity measure for DNNs based on the weights of trained networks applicable as an early stopping criterion. Rieck et al. (2019); Zhang et al. (2023) also show that topological properties can be useful for distinguishing models trained using different regularisers. Lacombe et al. (2021) extends neural persistence to take concrete input data into account, enabling out-of-distribution detection. Balwani & Krzyston (2022) use neural persistence to derive bounds on the number of units that may be pruned from trained neural networks. In this work we perform an in-depth analysis of neural persistence, confirming favourable properties of neural persistence for linear classifiers that is shown to not extend to DNNs. Watanabe & Yamana (2021; 2022) leverage persistent homology to detect overfitting in trained neural networks by considering subgraphs spanning across two layers. Different to this, we consider the global shape in network weights by proposing *deep graph persistence* which extends neural persistence to a complexity measure for DNNs. As our analysis shows that neural persistence is best applied to linear classifiers, we focus on creating a filtration that only implicitly incorporates intermediate nodes. This leads us to propose a filtration that is a special case of persistence diagrams of $\langle \phi = \lambda \rangle$-connected interactions due to Liu et al. (2020). In this case, instead of evaluating how neurons directly connected by linear transforms interact, we evaluate how input features interact with output units. Thereby, we reduce the layer-wise averaging of neural persistence to computing the neural persistence of a single summary matrix that contains minimum weight values on maximum paths from input units to output units.

## 3 Understanding neural persistence

In this section, we aim at providing a deeper understanding of neural persistence, which was introduced by Rieck et al. (2019). In particular, we identify *variance* of weights and *spatial concentration* of large weights as important factors that impact neural persistence. We formalise these insights by deriving tighter bounds on neural persistence in terms of max-over-row and max-over-column values of weight matrices.

Neural persistence is defined for undirected weighted graphs, and is used to compute the neural persistence for a layer. Before stating the definition of neural persistence, we recall the definition of complete bipartite graphs.

**Definition 3.1** (Bipartite graph)**.** A graph $G = (V, E)$ is called *bipartite* if vertices $V$ can be separated into two disjoint subsets $A, B$ with $V = A \cup B$ and $A \cap B = \emptyset$, and all edges $e \in E$ are of the form $e = (a, b)$ with $a \in A$ and $b \in B$, i.e. every edge connects a vertex in $A$ to one in $B$.

**Definition 3.2** (Complete bipartite)**.** $G$ is a *complete bipartite* graph if edges between all vertices in $A$ and all vertices in $B$ exist.

*Remark* 3.3. Note that any matrix $W \in \mathbb{R}^{n \times m}$ can be interpreted as the adjacency matrix of an undirected weighted complete bipartite graph. In this case, rows and columns correspond to vertices in $A$ and $B$, respectively. The matrix entries then resemble edge weights between all vertices in $A$ and $B$.

Rieck et al. assert that neural persistence can be defined in terms of the maximum spanning tree (MST) of a complete bipartite graph instead of persistent homology.

**Definition 3.4** (Neural persistence). Let $W \in [0;1]^{n \times m}$ be a matrix with $n$ rows, $m$ columns, and entries bounded below by 0 and above by 1. Throughout this paper, we denote entries in $W$ with row index $i$ and column index $j$ as $W_{i,j}$. As in Remark 3.3, $W$ can be interpreted as the adjacency matrix of an undirected weighted complete bipartite graph $G_W = (V_W, E_W)$. Let $\text{MST}(G_W) = (V_W, E_{\text{MST}(G_W)})$ with $E_{\text{MST}(G_W)} \subset E_W$ be the unique maximum spanning tree of $G_W$. In general, uniqueness is not guaranteed, but can be achieved by infinitesimal perturbations. Then, let $\text{MST}^w(G_W)$ be the set of weights for edges contained in the MST, i.e.

$$\text{MST}^w(G_W) := \{W_{v,v'} \mid (v, v') \in E_{\text{MST}(G_W)}\}. \tag{1}$$

The neural persistence $\text{NP}_p(W)$ is defined as

$$\text{NP}_p(W) := \left(1 + \sum_{w \in \text{MST}^w(G_W)} (1 - w)^p\right)^{\frac{1}{p}}, \tag{2}$$

and subsequently neural persistence for an entire neural network is defined as the average neural persistence across all individual layers. Weights, which can have arbitrary values, are mapped to the range $[0;1]$ by taking the absolute value and dividing by the largest absolute weight value in the neural network. Thus, for the remainder of this paper, we assume that all weights are in the range $[0;1]$. Also, we abbreviate neural persistence as "NP".

**Normalised neural persistence.** To make NP values of matrices with different sizes comparable, Rieck et al. propose to divide NP by the theoretical upper bound $(n + m - 1)^{\frac{1}{p}}$ (see Theorem 1 in (Rieck et al., 2019)). This normalisation maps NP values to the range $[0;1]$. We use this normalisation and $p = 2$ in all experiments, both the experiments concerning NP (Section 4) and also the experiments concerning DGP (Definition 5.1 and Appendix H.2). For caveats of this normalisation, refer to Appendix C.

**Bounds on neural persistence.** Defining NP in terms of the MST of a complete bipartite graph provides the interesting insight that the NP value is closely related to max-over-rows and max-over-columns values in the matrix. This characterisation provides intuitions about properties of weight matrices that influence NP. These intuitions are formalised in Theorem 3.5 as bounds on NP which are tighter than the bounds given in Theorem 2 in (Rieck et al., 2019).

**Theorem 3.5.** *Let $G_W = (V_W, E_W)$ be a weighted complete bipartite graph as in Definition 3.4 with edge weights given by $W$ and $V_W = A \cup B$, $A \cap B = \emptyset$. To simplify notation, we define*

$$\Phi_b := (1 - \max_{a \in A} W_{a,b})^p \quad \forall b \in B, \tag{3}$$

$$\Psi_a := (1 - \max_{b \in B} W_{a,b})^p \quad \forall a \in A. \tag{4}$$

*Using these shortcuts, we define*

$$L := \left(\sum_{b \in B} \Phi_b + \sum_{a \in A} \Psi_a\right)^{\frac{1}{p}}, \tag{5}$$

$$U := \left(|B \setminus B_{\not\prec A}| + \sum_{b \in B_{\not\prec A}} \Phi_b + \sum_{a \in A} \Psi_a\right)^{\frac{1}{p}}, \tag{6}$$

*where*

$$B_{\not\prec A} := \{b \in B \mid \forall a \in A : b \neq \operatorname*{argmax}_{b' \in B} W_{a,b'}\}. \tag{7}$$

$B_{\not\prec A} \subset B$ *can be thought of as the set of columns whose maximal element does not coincide with the maximal element in any row of $W$.*

*Then, the following inequalities hold:*

$$0 \leq L \leq \mathrm{NP}_p(W) \leq U \leq (n+m)^{\frac{1}{p}}. \tag{8}$$

*Proof (sketch).* For the lower bound, using properties of spanning trees, we construct a bijection between vertices $V$ (with one vertex excluded) and edges in $\mathrm{MST}(G_W)$. Each vertex $v$ is mapped to an edge that is connected to $v$. Using this bijection, we can bound the weight of each edge in the MST by the maximum weight of any edge connected to the respective vertex. Since maximum weights of edges connected to vertices correspond to max-over-rows and max-over-columns values, we obtain the formulation of $L$. For the upper bound, we observe that all max-over-rows and max-over columns values are necessarily included in $\mathrm{MST}^w(G_W)$. However, in some cases max-over-rows values and max-over-columns values coincide. Therefore, this observation leaves some values in $\mathrm{MST}^w(G_W)$ undetermined. For these, we choose the value that maximises NP, i.e. 0, to obtain an upper bound.

The detailed proof is provided in Appendix A and we also show in Appendix B.1 that the lower bound is generally tight and the upper bound tightens with increasing spatial concentration of large weights.

**Variance and spatial concentration of weights as factors that impact neural persistence.** The bounds on NP derived in Theorem 3.5 mostly depend on max-over-columns and max-over-rows values in $W$. Thus, they identify additional factors that impact NP, in particular the *variance* and *spatial concentration* of weights. The *spatial structure* of a matrix captures whether large entries concentrate on particular rows or columns, or only certain combinations of rows and columns. This intuition does not depend on any particular ordering of rows or columns because permuting them does not change whether large entries concentrate on some rows (or columns). However, detecting spatial concentration is a difficult problem in practice, because we need to simultaneously quantify what a large weight is, i.e. how much larger a weight is than the expected weight value, and how skewed the distribution of deviations of expected weight values in rows or columns is from the barycenter of the row- or column-wise weight distributions. In Appendix E, we discuss an attempt how to measure spatial concentration by sorting rows or columns by their mean value and then calculating a sortedness metric (Mannila, 1985) for the whole matrix. We then also discuss how to artificially construct matrices with a pre-defined level of spatial concentration. In Section 4, in contrast, we avoid quantifying spatial concentration explicitly, and instead rely on random permutation of weights, which destroys any spatial concentration that may have been present. In the following, we explain how the derived bounds identify these particular properties as factors that impact NP.

The lower bound $L$ is tighter when the variance of weights is smaller. In this case, it is more likely that the actual weight in the MST chosen for any vertex $v$ is close to the maximum weight connected to $v$. When mean weight values and variances of weights are highly correlated, which is the case in practise, NP increases with lower variance. The reason is that the lower variance causes the expected maximum value of a sample to be closer to the expected value, meaning the max-over-rows and max-over-columns values will also decrease together with lower mean and variance.

The upper bound $U$ is tighter when $|B_{\not\prec A}|$ is smaller. This is the case when large weights are concentrated on edges connected to few or even a single vertex, i.e. when there is relevant spatial concentration of large weights on certain rows or columns. In the extreme case, all edges with maximum weight for any vertex in $A$ are connected to the same vertex $b \in B$. Then, we know that edges with maximum weight, i.e. max-over-columns values, for all vertices in $B \setminus \{b\}$ will be part of the MST. In this case, we have equality of $\mathrm{NP}_p(W) = U$. Also, when large weights are concentrated on fewer rows or columns, max-over-rows and max-over-columns values will generally be lower, resulting in higher NP values.

## 4 Experimental analysis of neural persistence

### 4.1 Experimental setup

To study NP of neural networks and validate our insights in practise, we train a larg set of models. Namely, we train DNNs with exhaustive combinations of the following hyperparameters: Number of layers $\in \{2, 3, 4\}$, hidden size $\in \{50, 100, 250, 650\}$, and activation function $\in \{\tanh, \mathrm{relu}\}$. However, we do not apply dropout

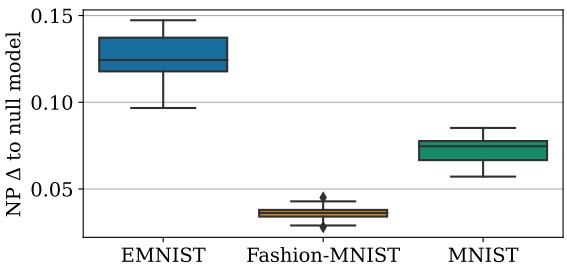
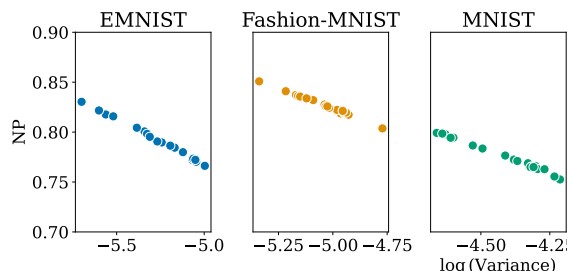

(a) Effect of randomly permuting weights in trained linear classifiers. Large changes in NP (NP$\Delta$) indicate the presence of spatial concentration of large weights. For each dataset, we train 20 models (see Section 4.1), which are fully characterised by the learned weight matrix.

(b) Relationship of log-variance of weights and NP conditioned on the respective dataset for linear classifiers. All models were trained for 40 epochs. The relationship is almost perfectly linear. Linear regression fits give $R^2 \approx 99\%$ for all datasets.

Figure 1: Analysing the relation between neural persistence and spatial concentration (left) and variance of weights in linear classifiers.

or batch normalisation. Note, that in our terminology a "layer" refers to a linear transform, i.e. weight matrix.

We use the Adam optimizer (Kingma & Ba, 2015) with the same hyperparameters as Rieck et al., i.e. with a learning rate of 0.003, no weight decay, $\beta_1 = 0.9$, $\beta_2 = 0.999$, and $\epsilon = 10^{-8}$. Each model is trained for 40 epochs with batch size 32, and we keep a checkpoint after every quarter epoch. We train models on three datasets, namely MNIST (LeCun et al., 1998), EMNIST (Cohen et al., 2017), and Fashion-MNIST (Xiao et al., 2017). For EMNIST, we use the *balanced* setting. Rieck et al. do not train models on EMNIST, but we decide to include this dataset, because it is more challenging as it contains more classes, namely 49 instead of 10 in the case of MNIST and Fashion-MNIST. For each combination of hyperparameters and dataset, we train 20 models, each with different initialisation and minibatch trajectory. If not stated otherwise, we analyse the models after training for 40 epochs. Additionally, we train 20 linear classifiers (perceptrons) for each dataset using the same hyperparameters (optimizer, batch size, number of epochs) as for deep models.

## 4.2 Neural persistence for linear classifiers

NP is defined for individual layers in Definition 3.4. Generalisation to deeper networks is achieved by averaging NP values across different layers. Here, we first analyse NP for models with only one layer, i.e. for linear classifiers. In this case, there are no effects of averaging or different matrix sizes.

For linear classifiers, we show the presence of spatial structure trained models. Furthermore, we demonstrate the correspondence of NP and variance, after controlling for spatial structure. We find that both factors, variance of weights and spatial structure of large weights, vary in trained linear classifiers. Therefore, because these two factors can already explain a significant amount of the variation of NP values according to our insights in Section 3, NP is likely to exhaust its full capabilities in the case of linear classifiers.

**Spatial structure.** We demonstrate that weights of trained linear classifiers exhibit spatial structure in the following way: We compare the NP of actual weight matrices to suitable null models. As null models, we choose randomly shuffled weight matrices, i.e. we take actual weight matrices of trained linear classifiers and randomly reassign row and column indices of individual weight values without changing the set of weights present in the matrix. This intervention destroys any spatial structure in the weights, but preserves properties that are independent of the spatial distribution, such as variance.

Figure 1a shows that weight matrices of linear classifiers trained on all datasets used in this paper, namely MNIST, EMNIST and Fashion-MNIST, exhibit relevant spatial concentration of large weights. Random permutation of weights leads to large changes in NP values. For comparison, the mean absolute deviation of NP from the mean for different initialisations is generally less than 0.02. In addition, we conduct two-sample Kolmogorov-Smirnov (KS-) tests to test if the distribution of neural persistence values of the layer

without permutation is different from the distribution of neural persistence values after permutation. Indeed, all KS-tests reject the null hypothesis that neural persistence values with and without permutation are identically distributed with p-values $\ll 10^{-6}$. Changes resulting from random permutation are smallest for models trained on Fashion-MNIST. Possibly, visual structures in Fashion-MNIST data, i.e. clothes, are more complex and spatially distributed compared to characters or digits in MNIST and EMNIST which means that more different input features will receive large weights.

**Variance of weights.** Having established the presence of spatial concentration of large weights in weight matrices of trained linear classifiers, we analyse the relationship of NP and the variance of weights. Figure 1b shows that a linear relationship between NP and the variance of weights can be observed for trained linear classifiers. In particular, the $R^2$ score of a linear regression fit is $\approx 99\%$ for all datasets. This is expected, since all else being equal, the dataset is the only factor that impacts the spatial concentration of large weights in linear classifiers. Recall that in linear classifiers, large weights identify important features in the input data. Therefore, multiple linear classifiers trained on the same dataset will generally assign large weights to the same entries in the weight matrix. Eliminating this factor by separating models trained on different datasets in the analysis leaves variance as the only factor identified in Section 3 that impacts NP.

### 4.3 Neural persistence for deep neural networks

In Section 4.2, we established that trained linear classifiers exhibit spatial concentration of large weights and that, conditioned on the spatial concentration of large weights, NP responds linearly to changes in variance of weights. Naturally, we are interested in whether this extends to DNNs. Motivated by our theoretical analysis, we analyse the impact of the variance of weights and the spatial concentration of large weights on NP. In particular, we find that no relevant spatial structures is present in later layers of DNNs, and therefore NP corresponds roughly linearly to the variance of weights, as effects of spatial structure become irrelevant.

**Spatial structure.** To analyse the spatial structure in DNNs, we repeat the permutation experiment from Section 4.2, where we shuffle entries in weight matrices and compare the resulting NP values to NP values for the original weight matrices of trained neural networks. The resulting changes in NP values are visualised in Figure 2, confirming that irrespective of the dataset or number of layers, the NP values of later layers in the network are insensitive to the random permutation of entries. The changes in NP values are mostly less than 0.02, which is a good bound for the variation of NP values resulting from different initialisations and minibatch trajectories. Additionally, we run KS-tests for each hyperparameter combination (i.e. activation function, number of layers, hidden size, dataset) and layer in the respective network to test if the neural persistence values with and without permutation are distributed differently. This results in a total of 216 tests. The KS-tests fail to reject the null hypothesis that the neural persistence values with and without permutation are identically distributed (p-value $> 0.05$) for all layers not connected to the input features. For the first layer that is connected to the input features, the KS-tests reject the null hypothesis in $\frac{21}{24}$ of cases for EMNIST, in $\frac{19}{24}$ cases for Fashion-MNIST, and in $\frac{20}{24}$ cases for MNIST. These results are in agreement with Figure 2. These findings indicate the absence of any spatial structure in the large weights of trained neural networks that is relevant for NP in later layers of trained neural networks.

**Variance of weights.** Unlike spatial structure, differences in the variance of trained weights also exist in the case of DNNs. Therefore, our results so far suggest that, in the absence of relevant spatial structure, the variance of weights becomes the main factor that corresponds to changes in NP. Indeed, in Figure 3 we again observe a roughly linear correspondence of NP with the log-transformed global variance of all weights in the trained network.

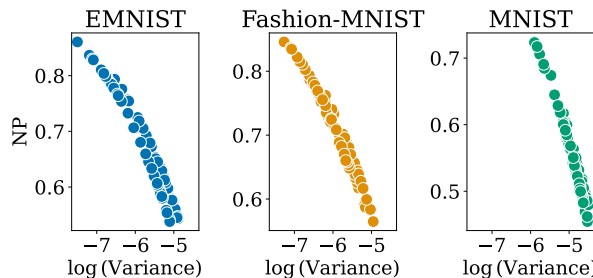

Figure 3: Roughly linear correspondence of log-variance of weights and NP for DNNs (3 or 4 layers, hidden size $\in \{250, 650\}$).

As is the case for linear classifiers, matrix sizes influence the effective NP values. However, the matrix sizes that influence the final NP value depend on the dataset (input and output layer), the hidden size,

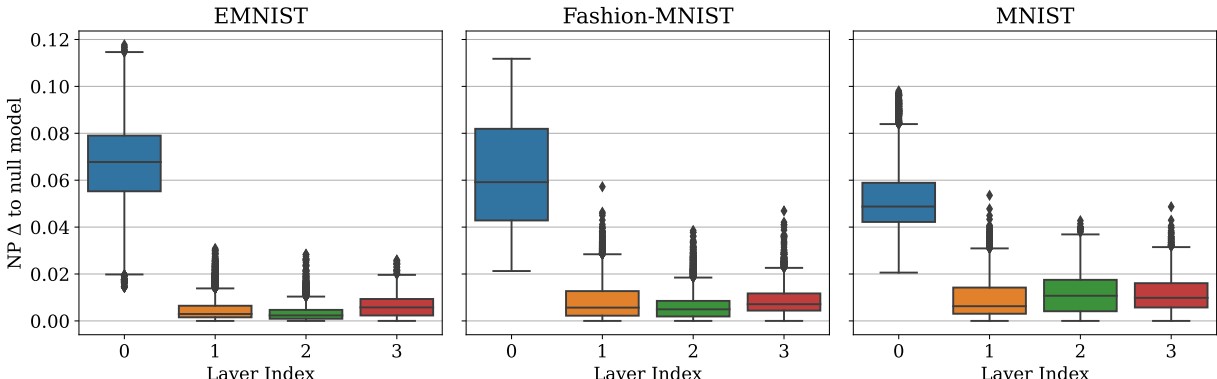

Figure 2: Effect of randomly permuting weights in weight matrices in trained DNNs. Large changes in NP indicate the presence of spatial concentration of large weights. We factorise the results by dataset and index of the layer for which neural persistence is computed.

and the number of layers. Therefore, we only include DNNs with 3 or 4 layers and hidden size $\in \{250, 650\}$ in Figure 3. Additional supporting results are provided in Appendix H.1.

The absence of spatial structure in layers that are located later in the network along with the strong correspondence of NP with the variance of weights naturally raises questions about the additional information provided by NP that goes beyond the variance of weights. We argue that little additional information is provided: Rieck et al. suggest to use the growth of NP values for early stopping. This is very useful in cases where no validation data is available. Therefore, we compare the variance as early stopping criterion to NP.

In Appendix H.2, we show that using the variance of weights in place of NP as early stopping criterion yields similar results: Using the variance as criterion leads to training for 2 more epochs on average compared to NP, with small increase in the resulting test set accuracy. Additionally, we find that the variance of weights yields similar results as NP in distinguishing models trained with different hyperparameters, e.g. dropout or batch normalisation. Further details are given in Appendix H.3.

## 5 An improved complexity measure for deep feed-forward neural networks

Our analysis in Sections 3 and 4.2 demonstrated that the variance of the weights dominates the neural persistence measure. Moreover, variance proves to be equally effective as the computationally demanding NP when used as an early stopping criterion.

In this section, we introduce our *deep graph persistence* (DGP) measure with the intention of reducing the dependency on the variance of network weights, and instead increasing dependency on the network structure. Our DGP alters the filtration used in the calculation of neural persistence. This addresses underlying issues in neural persistence, and we show that DGP compares favourably to related methods on the downstream task of covariate shift detection.

### 5.1 Definition of DGP

Our proposed DGP measure addresses two caveats of neural persistence: (1) Neural persistence of a layer is highly correlated with the variance of the respective weight matrix; (2) Neural persistence of a network is computed by averaging layer-wise values.

To address the first caveat, we standardise the weight matrices in each layer before normalisation. This mitigates effects of layer imbalances (Du et al., 2018), i.e. all weights appearing very small because one entry in a particular layer is very large. We standardise the weights by subtracting the layer mean $\mu^l$ and dividing by the standard deviation $\sigma^l$ of weights in the same layer:

$$\overline{W}^l_{i,j} = \frac{W_{i,j} - \mu^l}{\sigma^l}. \tag{9}$$

We then follow the normalisation procedure of neural persistence by taking the absolute values of the normalised weights $\overline{W}_{i,j}^{l}$ and dividing by the largest resulting value in the network $\overline{W}_{\max}$. We can then define the set $\overline{W}_G := \left\{ |\overline{W}_n|/\overline{W}_{\max} \forall n = 0, \dots, N^w \right\}$, indexed in descending order. Here, $N^w$ is the cardinality of the set $\overline{W}_G$, i.e. the number of parameters in the full network.

To address the second caveat, we consider the whole computational graph at once when computing DGP similar to (Liu et al., 2020) and (Gebhart et al., 2019). This allows including structural properties between layers which is different from the averaging over individual layers in neural persistence. The filtration proposed by Rieck et al. for neural persistence is computed layer-wise, and within each layer nodes are considered as independent components. Specifically, they include an edge in the calculation of neural persistence, i.e. in the MST, in the order defined by decreasing magnitude whenever it creates a new connected component consisting of input nodes and output nodes of the respective layer. We extend this intuition to the entire neural network instead of treating layers independently. We add edges to the computational graph sorted by decreasing (absolute) magnitude. Whenever an edge creates a new connected component of input and output nodes of the full neural network, we include this edge. Finally, we compute DGP similar to neural persistence, but by using the previously described set of edges for the full neural network, instead of using edges from individual layers only. Note, that this intuition corresponds to the filtration induced by definition of $\langle \phi = \lambda \rangle$-connected by Liu et al. (2020) (Definition 3 in their paper). The Kruskal algorithm used to construct the MST also keeps track of the connected components, just like $\langle \phi = \lambda \rangle$-connected does. However, the definition of $\langle \phi = \lambda \rangle$-connected is broader, as the filtration is also applied to other layers than the input layer and considers connected components with more than one element. We point out that this is different from our approach, because in our case a singleton connected components dies once it is merged with another connected component. In summary, the feature-interaction approach of Liu et al. provide another motivation to extend NP to capture interactions between all layers.

In essence, we adapt the calculation of neural persistence to involve just the input nodes $V_{\text{in}}$ and output nodes $V_{\text{out}}$ of $G$ in a way that respects persistent *paths* through the entire network. Mathematically, this is equivalent to calculating the MST of a complete bipartite graph $\widehat{G} := (\widehat{V}, \widehat{E})$, with the set of vertices $\widehat{V} := V_{\text{in}} \cup V_{\text{out}}$ representing the input and output nodes of $G$ and a set of edges $\widehat{E}$ between these nodes.

In order to assign weights to the edges in $\widehat{E}$, we turn to the full network graph $G = (V, E)$. For any choice of input node $a \in V_{\text{in}}$ and output node $b \in V_{\text{out}}$, we consider the set of directed paths $P_{ab}$ (following the computational direction of the network) from $a$ to $b$. Each path $p \in P_{ab}$ contains $L$ edges, where $L$ is the number of hidden layers in the network, and the edges are in different layers. Let $\vartheta : E \to \overline{W}_G$ denote the mapping that returns the edge weight of any edge $(a, b) \in E$. We can now define $\Theta : \widehat{E} \to \overline{W}_G$ to compute the edge weight of any edge $(\widehat{a}, \widehat{b}) \in \widehat{E}$ as

$$\Theta(\widehat{a}, \widehat{b}) := \max_{p \in P_{ab}} \min_{(u,v) \in p} \vartheta(u, v). \tag{10}$$

Thus, $\Theta$ assigns the filtration threshold to $(\widehat{a}, \widehat{b})$ for which there exists at least one directed path between $a$ and $b$ in the full network graph $G$. The resulting persistence diagram for DGP is calculated from the MST of the bipartite graph corresponding to

$$S_{\widehat{ab}} = \Theta(\widehat{a}, \widehat{b}). \tag{11}$$

Different to neural persistence, this approach encodes information about connectivity in the full network. Finally, we can define the DGP measure in an analogous fashion to neural persistence:

**Definition 5.1** (Deep graph persistence). Let $S \in [0; 1]^{|V_{\text{in}}| \times |V_{\text{out}}|}$ be a summarisation matrix with $|V_{\text{in}}|$ rows, $|V_{\text{out}}|$ columns, and entries given by $\Theta$. The deep graph persistence $\text{DGP}_p$ of the full network $G$ is defined as

$$\text{DGP}_p(G) := \left( 1 + \sum_{w \in \text{MST}^w(\widehat{G})} (1 - w)^p \right)^{\frac{1}{p}}. \tag{12}$$

This formulation requires calculating the MST for just a single matrix $S$ rather than for every layer. Furthermore, $S$ has a fixed dimensionality regardless of architectural choices, such as layer sizes, and it can be efficiently constructed by dynamic programming (see Algorithm 1 in Appendix H). Concretely, the computational complexity for constructing $S$ is $\mathcal{O}(L \cdot o \cdot d^2)$, where $L$ is the number of linear transforms in the network, $o$ is the number of output units, and $d$ is an upper bound for the number of input units and the hidden size. The computational complexity of calculating the MST of $S$ using Kruskal's algorithm is $\mathcal{O}(V^2 \cdot \log V)$, where $V$ is the number of input features plus the number of output units.

We show in Appendix H.4 that DGP indeed does not exhibit the correspondence to the variance of weights that we observed for NP. Also, we check if DGP yields a better early stopping criterion than NP or the variance of weights. However, our results in Appendix H.2 are negative. There is no relevant practical difference between NP and DGP when used as early stopping criterion. Since Rieck et al. only provide empirical, and no theoretical, justification for why NP can be used as early stopping criterion, we conclude from our findings that assessing generalisation or even convergence only from a model's parameters without using any data remains difficult. While related methods also try to avoid labelled data as early stopping criterion, they always require access to information based on model computations, such as gradients (Mahsereci et al., 2017; Forouzesh & Thiran, 2021) or node activations (Corneanu et al., 2020).

Therefore, in the following, we focus on the utility for downstream applications which we see as a more fruitful direction for applying methods related to NP.

### 5.2 Sample-weighted deep graph persistence for covariate shift detection

Lacombe et al. (2021) have shown the usefulness of the neural persistence approach for covariate shift detection by including individual data samples in the calculations of Topological Uncertainty (TU). Concretely, TU uses the same method as NP, namely calculating the 0-order persistence diagrams of individual layers (i.e. the MST of the corresponding bipartite graph) and then averaging scores across layers. In contrast to NP, TU is applied to the *activation graph* (defined below) of a neural network (Gebhart et al., 2019).

To use our proposed deep graph persistence in downstream tasks that require access to data, we introduce the notion of *sample-weighted deep graph persistence*. For this, we calculate DGP on weights multiplied with the incoming activation that is obtained when applying the network to a given sample, instead of the static weights of the trained model. In the following, we formally define sample-weighted deep graph persistence.

**Definition 5.2** (Activation graph)**.** For a neural network $G$, the activation graph is defined as the input-induced weighted graph $G(x)$ obtained as follows. For an edge connecting an input node of layer $\ell$ with an output node of layer $\ell$ in $G$, the edge weight is given by the activation value of the input node multiplied by the corresponding element in the weight matrix, i.e. $W_{i,j}^\ell \cdot x_i^\ell$. Following Lacombe et al. (2021), we use the absolute value $|W_{i,j}^\ell \cdot x_i^\ell|$.

**Definition 5.3** (Sample-weighted deep graph persistence)**.** For a neural network $G$ and a sample $x$, we first obtain the activation graph $G(x)$. We then calculate the summarisation matrix $S(x)$ using the edge values of the activation graph instead of the edge weights of the non-sample weighted computational graph.

We use our sample-weighted DGP for the detection of covariate shifts, i.e. shifts in the feature distribution in relation to the training or in-domain data. This task was also considered in TU (Lacombe et al., 2021) which can be considered as an application of NP. We show that modifying neural persistence to mitigate its identified caveats improves performance of derived methods in downstream applications. To compare DGP and TU, we build on work by Rabanser et al. (2019), who have established a method to detect covariate and label shifts by leveraging representations from trained neural networks.

**Method.** The method proposed by Rabanser et al. (2019) attempts to detect if a batch of data samples contains out-of-domain data by comparing it to in-domain data. The method assumes the availability of (a limited amount of) in-domain data. Concretely, their proposed method first extracts representations of the in-domain and test data from a trained neural network. This results in two feature matrices $F_{\text{clean}} \in \mathbb{R}^{n \times d}$ and $F_{\text{test}} \in \mathbb{R}^{m \times d}$, where $n$ is the number of in-domain datapoints, $m$ is the number of test datapoints (here, we assume $n = m$), and $d$ is the number of features in the representation extracted from the trained model.

For each feature dimension $i \in \{1, \ldots, d\}$, a two-sample Kolmogorov-Smirnov (KS) test is conducted to test the hypothesis that the feature values in feature dimension $i$ are sampled from the same underlying distribution in $F_{\text{test}}$ and $F_{\text{clean}}$. Using Bonferroni correction when $d$ tests are performed, the null hypothesis, namely that the underlying distributions are equal, is rejected at a confidence level of 95% if the $p$-value of the KS test is smaller than $0.05 \cdot \frac{1}{d}$. Rabanser et al. conclude that $F_{\text{test}}$ contains out-of-domain datapoints if at least one of the $d$ tests fails.

In order to use deep graph persistence and TU in this setting, we use the weights in the MST used for calculating the DGP value (Definition 5.1) or the NP value (Definition 3.4). This is equivalent to using the persistence diagram used for calculating DGP (or NP). To ensure consistency with the persistence diagram formulation (see (Lacombe et al., 2021, Sec. 2.2)), the set of weights in the MST is sorted by weight magnitude and represented as a vector. For a network with $k$ input nodes and $c$ output nodes corresponding to the $c$ classes, this representation yields feature vectors of dimension $k+c-1$. The $-1$ is due to the MST containing one edge less than there are nodes in the graph. Following Lacombe et al. (2021), we calculate the differences to class-wise mean representation. Class-wise mean representations are defined as follows: Let $f(x_i)$ be a representation for sample $x_i$ (of $k$ given samples) extracted from the trained model, i.e. the sorted set of weights in the MST of a layer in the activation graph. Let $l_j$ the number of samples with class $j$ and $y_i$ the class of $x_i$. Then, the class-wise mean representation $\mu_j$ is defined as

$$\mu_j = \frac{1}{l_j} \sum_{i=1}^{k} \delta[y_i = j] \cdot f(x_i). \tag{13}$$

Given $k$ labeled training samples $\{\mathbf{x}_i, y_i\}_{i=1,\ldots l}$ (in our case, $l = 1000$) and feature extraction function $f : \mathbb{R}^k \to \mathbb{R}^d$ that extracts MST weights (or other representations such as the vector of softmax outputs) the difference of the representation of a sample $x_{\text{test}}$ to the mean representation of class $j$ is given by

$$\Delta_j(x_{\text{test}}) = \|f(x_{\text{text}}) - \mu_j\|_2, \tag{14}$$

where $l_j$ is the number of training samples in class $j$. Lacombe et al. (2021) use the difference to the mean representations only of the predicted class for sample $x_{\text{test}}$ as a scalar score (averaged across layers). However, we find that using the $c$-dimensional vector which contains the differences to mean representations for training samples of all classes as done in (Hensel et al., 2023) to be more effective for both TU and DGP:

$$\Delta(x_{\text{test}}) = \begin{pmatrix} \Delta_1(x_{\text{test}}) & \Delta_2(x_{\text{test}}) & \ldots & \Delta_c(x_{\text{test}}) \end{pmatrix}. \tag{15}$$

**Baselines.** We compare DGP to three baselines for covariate shift detection. **Softmax** outputs of the model are used by the seminal work of Rabanser et al. (2019). Topological Uncertainty (**TU**) (Lacombe et al., 2021) effectively calculates NP on the activation graph (Gebhart et al., 2019). The recently proposed **MAGDiff** (Hensel et al., 2023) uses the complete final layer of the activation graph for sample $x$ as extracted representation $f(x)$ to detect corrupted images. TU and MAGDiff do not use features directly, but their difference to the mean features of training samples grouped by classes. We compare to TU, as it directly uses neural persistence, and we also incorporate other methods as baselines. We use the evaluation protocol proposed by Rabanser et al..

**Setup.** To evaluate the different methods on the covariate shift detection task, we train MLP models on MNIST, Fashion-MNIST and CIFAR-10. For each dataset, we train models with all combinations of hidden size $\in \{100, 650\}$ and number of layers $\in \{2, 4\}$. For each hyperparameter combination, we train 5 models with different initialisations. Hyperparameters and training setup are as described in Section 4.1.

**Evaluation.** Covariate shifts are simulated on synthetic data by artificially corrupting in-domain data with added noise or other augmentations as proposed by Rabanser et al. (2019). In our evaluation, we corrupt images from MNIST, Fashion-MNIST and CIFAR-10 by adding Gaussian noise (i.e. noise sampled from a Gaussian distribution), uniform noise (i.e. noise sampled from a uniform distribution), applying input dropout (i.e. changing the colour or brightness of pixels to black), or applying Gaussian blur to images. All corruptions are applied in varying strengths. Details regarding the strength of corruptions are in Appendix G.

We report the detection ratio (in percent) which measures the frequency of detecting corrupted samples for each method. Specifically, it captures how often each method returns feature representations which make

| | Dataset | CIFAR-10 | | | | Fashion-MNIST | | | | MNIST | | | |
|---|---|---|---|---|---|---|---|---|---|---|---|---|---|
| $n$ | Corruption | GB | GN | PD | UN | GB | GN | PD | UN | GB | GN | PD | UN |
| 10 | Softmax | **2.22** | 2.84 | 4.76 | 1.97 | **1.76** | **5.78** | 5.55 | **3.04** | 2.11 | 15.30 | **19.67** | 7.44 |
| | TU | 1.32 | 3.66 | 16.31 | 1.34 | *1.44 | 1.79 | 7.55 | 1.54 | 1.63 | 5.07 | *17.03 | 2.27 |
| | MAGDiff | 1.83 | 5.24 | 9.86 | 1.56 | 1.52 | 3.42 | 7.07 | 2.16 | **2.39** | 8.48 | 19.49 | 4.60 |
| | DGP (Ours) | *1.23 | *9.18 | *26.17 | *2.53 | 1.44 | *4.10 | *12.76 | *2.87 | *1.91 | *25.79 | 14.90 | *16.96 |
| 20 | Softmax | 4.56 | 6.71 | 14.31 | 3.79 | 4.00 | 16.44 | 14.33 | 9.20 | 4.14 | 32.60 | **36.94** | 20.17 |
| | TU | *3.17 | 10.91 | 29.67 | 3.12 | *3.04 | 4.22 | 18.28 | 3.45 | 3.85 | 13.85 | *31.02 | 6.54 |
| | MAGDiff | **5.11** | 14.48 | 22.42 | 3.62 | 3.20 | 10.51 | 17.83 | 5.78 | **5.47** | 20.72 | 36.14 | 12.98 |
| | DGP (Ours) | 2.84 | *22.65 | *43.51 | *8.41 | 2.92 | *12.53 | *25.74 | *8.75 | *4.92 | *43.57 | 28.44 | *32.56 |
| 50 | Softmax | **6.16** | 10.26 | 23.53 | 3.24 | **4.37** | 28.64 | 23.31 | 17.42 | 4.20 | 48.69 | **51.30** | 32.00 |
| | TU | 4.11 | 20.08 | 44.14 | 4.91 | *3.59 | 7.20 | 29.82 | 4.08 | 6.34 | 24.02 | *46.74 | 12.28 |
| | MAGDiff | 9.38 | 25.51 | 35.95 | 6.80 | 3.30 | 19.49 | 29.34 | 11.13 | 8.24 | 35.22 | 50.58 | 23.70 |
| | DGP (Ours) | *4.61 | *36.33 | *60.49 | *16.57 | 3.09 | *25.78 | *39.72 | *18.84 | *9.71 | *62.13 | 43.20 | *48.92 |
| 100 | Softmax | 11.84 | 18.91 | 36.57 | 5.28 | **9.61** | **43.06** | 35.44 | 28.97 | 7.23 | 65.03 | **66.84** | 47.75 |
| | TU | 8.79 | 31.31 | 58.81 | 9.98 | *7.08 | 16.07 | 43.47 | 8.53 | 12.35 | 37.17 | *62.33 | 21.31 |
| | MAGDiff | **17.79** | 38.49 | 50.47 | 13.35 | 5.96 | 32.84 | 42.70 | 21.07 | 15.55 | 51.31 | 65.96 | 38.59 |
| | DGP (Ours) | *10.78 | *50.91 | *79.88 | *27.70 | 6.48 | *40.74 | *54.41 | *31.78 | *20.16 | *79.21 | 58.14 | *64.71 |
| 200 | Softmax | 18.77 | 27.71 | 48.20 | 8.37 | **16.29** | **56.78** | 46.94 | 41.12 | 10.60 | 79.23 | **78.90** | 61.45 |
| | TU | 15.90 | 42.90 | 71.13 | 16.81 | 12.55 | 26.72 | 56.61 | 16.05 | 19.38 | 50.64 | *75.64 | 31.92 |
| | MAGDiff | **27.58** | 50.97 | 63.80 | 20.76 | 10.45 | 45.90 | 55.58 | 32.82 | 23.15 | 65.83 | 78.13 | 52.32 |
| | DGP (Ours) | *19.67 | *64.36 | *89.39 | *39.16 | *13.11 | *55.58 | *67.51 | *45.22 | *30.19 | *91.08 | 71.58 | *78.71 |

Table 1: Covariate shift detection results for various corruptions and datasets. Scores are given in percent, i.e. in the range $[0; 100]$. GB means Gaussian blur, GN means Gaussian noise (i.e. noise sampled from a Gaussian distribution), PD means pixel dropout, and UN means uniform noise. Bold entries denote the best result among methods for each combination of $n$, dataset and corruption type. With the asterisk, we mark whether DGP or TU achieves a better score for the respective combination.

the KS tests reject the hypothesis that the underlying feature distributions are equal for in-domain and corrupted images. A score of 100 means that 100% of samples with corrupted images have been detected.

**Results.** In Table 1, we show covariate shift detection ratios on the CIFAR-10, Fashion-MNIST, and MNIST datasets for Gaussian blur, Gaussian noise, pixel dropout, and uniform noise. Interestingly, however, DGP does not manage to detect more subtle image transformations, such as rotation and zoom, equally well as the baselines, so here we focus on noise-type corruptions and leave exploring other types of shifts to future work. Also in Table 1, we show results for different features extraction methods and number of samples $n$ used for the detection. We observe that DGP performs better than baselines on most combinations of datasets and corruptions, i.e. DGP detects corruptions more often and needs fewer samples to detect corruptions. DGP gives particularly strong results on CIFAR-10. There, DGP surpasses the baselines on all corruption types except for Gaussian blur, where it still performs better than TU. Results on MNIST are similar. On MNIST, DGP is stronger for Gaussian blur detection for larger sample sizes, but fails to detect pixel dropout similar to the baselines. Results on Fashion-MNIST are more mixed. Here, DGP only performs better than the baselines on detecting uniform noise for larger sample sizes, and Softmax remains superior on detecting Gaussian noise and Gaussian blur, but with small margin. In both cases, DGP performs better than TU.

DGP yields improvements over TU for all corruptions and datasets with the exception of detecting pixel dropout on MNIST and Gaussian blur on Fashion-MNIST, where DGP is outperformed by TU. The difference in these cases is relatively small compared to the improvements DGP achieves in other settings.

## 5.3 Ablation study on sample-weighted deep graph persistence

In Table 2, we show the effect of different components in the calculation of DGP on the covariate shift detection performance. In particular, we analyse the impact of standardising layers by substracting the layer-wise mean and of dividing by the layer-wise standard deviation. "TU (+ norm)" refers to TU, but removing mean and standard deviation of layers of the activation graph before calculating representations. "DGP (-norm)" refers to DGP, but without the removal of mean and standard deviation before constructing the matrix $S$. Due to the high computational cost of calculating TU for many samples, we provide results for the MNIST dataset only. Table 2 shows that both modifications, i.e. removing the averaging of layer-

| $n$ | 20 | | | | 50 | | | | 100 | | | |
|---|---|---|---|---|---|---|---|---|---|---|---|---|
| Corruption | GB | GN | PD | UN | GB | GN | PD | UN | GB | GN | PD | UN |
| TU | 3.85 | 13.85 | **31.02** | 6.54 | 6.34 | 24.02 | **46.74** | 12.28 | 12.35 | 37.17 | **62.33** | 21.31 |
| TU (+ norm) | **4.94** | 40.55 | 28.51 | 28.34 | 9.31 | 57.44 | 43.40 | 43.23 | 18.48 | 75.06 | 58.26 | 57.80 |
| DGP (- norm) | 3.66 | 35.15 | 27.72 | 22.46 | 5.15 | 51.78 | 41.15 | 35.70 | 11.26 | 67.71 | 55.95 | 49.74 |
| DGP | 4.92 | **43.57** | 28.44 | **32.56** | **9.71** | **62.13** | 43.20 | **48.92** | **20.16** | **79.21** | 58.14 | **64.71** |

Table 2: Ablation results on MNIST. Scores are corruption detection ratios (in %). "norm" denotes applying layer-wise standardisation. The full table with results for $n = 10$ and $n = 200$ is in Appendix H.6.

wise scores (DGP (- norm)) and standardisation (TU (+ norm)), already improve performance individually. Standardisation alone generally leads to stronger performance than DGP without standardisation. Together, they yield further albeit smaller improvements. Note, that again we observe a weaker performance of DGP for pixel dropout. In summary, this suggests that our proposed modifications indeed target properties of NP that so far impede its usefulness for downstream tasks.

## 6 Conclusion

In this work, we thoroughly analysed the neural persistence measure and found both theoretically and empirically that the variance of weights and spatial concentration of large weights are main factors that impact neural persistence. We compared the behaviour of neural persistence for linear classifiers and for deep feed-forward neural networks. For trained linear classifiers, we observed that the variance of weights varies for different independent training runs and that the spatial concentration or large weights varies for different datasets. However, we find that later layers in deep feed-forward neural networks do not exhibit relevant spatial structure. We conclude that neural persistence is a more relevant metric for linear classifiers than for deep feed-forward neural networks, because in the latter case we find neural persistence to provide little information beyond the variance of weights. As neural persistence is more costly to compute than the variance of weights without yielding additional insights, we hypothesise that a reformulation or different application of neural persistence is necessary for deep neural networks. This leads us to propose deep graph persistence as an extension of neural persistence. Deep graph persistence directly addresses the caveats of neural persistence. Finally, we demonstrate the utility of deep graph persistence for covariate shift detection, which permits direct comparison to previous work in applying topological methods in downstream tasks.

## Acknowledgements

This work was supported by DFG project number 276693517, BMBF FKZ: 01IS18039A, ERC (853489 - DEXIM), EXC number 2064/1 – project number 390727645. Ole Winther is supported by the Novo Nordisk Foundation (NNF20OC0062606) and the Pioneer Centre for AI, DNRF grant number P1. Anders Christensen thanks the ELLIS PhD program for support.

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

# A  Proof of Theorem 3.5

Before we give the full argument proving the bounds in Theorem 3.5, we recall relevant definitions from graph theory.

**Definition A.1** (Spanning tree)**.** A spanning tree of an undirected graph $G = (V, E)$ is a connected acyclic subgraph of $G$ that contains every vertex (Sections 3-7 in (Deo, 1974)).

*Remark* A.2. A relevant property of spanning trees, and trees in general, is that for all vertices $v, v'$, there is a unique path from $v$ to $v'$ (Theorem 3-1 in (Deo, 1974)).

**Definition A.3** (Maximum spanning tree)**.** The maximum spanning tree $\mathrm{MST}(G)$ of a weighted graph $G = (V, E)$ is a spanning tree with maximal (summed) edge weights (Theorem 3-16 in (Deo, 1974). If all weight values are unique, which we assume here, the maximum spanning tree is unique as well. We denote the maximum spanning tree by $\mathrm{MST}(G) = (V, E_{\mathrm{MST}})$.

*Remark* A.4. The maximum spanning tree $\mathrm{MST}(G)$ of graph $G$, and any tree in general, contains exactly $|V| - 1$ edges (Theorem 3-3 in (Deo, 1974)).

**Definition A.5** (Rooted tree)**.** A rooted tree is a tree where exactly one vertex $u \in V$ is singled out and called "root" (Section 3-5 in (Deo, 1974)). Any (unrooted) tree can be turned into a rooted tree by picking an arbitrary vertex as root.

**Definition A.6** (Arborescence)**.** An arborescence is a rooted tree with directed edges, where all edges point away from the root (Section 9-6 in (Deo, 1974)).

*Remark* A.7. An inductive argument over the path length shows that any (undirected) rooted tree can be turned into an arborescence by inducing the natural orientation on edges away from the root.

To see this, consider a node $v$ where the unique path from the root $u$ to $v$ has length $l$, i.e. contains $l$ edges. The case $l = 1$ is clear, because the respective edges are directed from the root to its neighbours. Now, consider the case $l > 1$. Then, there is a node $v'$ on the path from $u$ to $v$ which is a neighbour of $v$, i.e. the path length from $u$ to $v'$ is $l - 1$. By the induction hypothesis, the path from $u$ to $v'$ is directed and points away from the root. Similarly, the induction step is completed as the path from $v'$ to $v$ is directed and points from $v'$ to $v$. In conclusion, we see that any subtree with maximal path length $l$ can be turned into an arborescence (since the choice of $v$ is unrestricted), and thus the same is true for the entire tree.

*Remark* A.8. In an arborescence, every vertex $v$ except for the root $u$ has in-degree one, i.e. exactly one (directed) edge points towards $v$. The root $u$ has in-degree zero, i.e. no (directed) edge points towards $u$ (Theorem 9-2 in (Deo, 1974)).

*Remark* A.9. Let $T = (V_T, E_T)$ be a rooted tree with root $u \in V_T$. Through Remark A.8, we get a bijection $\varphi : V_T \setminus \{u\} \to E_T$ that maps every non-root vertex to its unique incoming edge. $\varphi$ is well defined because of Remark A.8. Also, $\varphi$ is injective, because every edge $e \in E_T$ has one and only one vertex that $e$ points to. Finally, $\varphi$ is surjective, because for every edge $e \in E_T$ there is a vertex that $e$ points to.

*Remark* A.10. Let $T = (V_T, E_T)$ be a rooted tree with root $u \in V_T$ and weighted edges. Let $\psi : E_T \to \mathbb{R}$ be the function that assigns weights to edges of $T$, i.e. $\psi$ maps edges to their corresponding weights. Then, we can extend $\varphi$ from Remark A.9 to $\varphi_w : V_T \setminus \{u\} \to \mathbb{R}$ which maps a vertex $v \in V_T \setminus \{u\}$ to the weight of the unique edges $e_v \in E_T$ that points towards $v$, i.e. $\varphi_w = \psi \circ \varphi$.

**Definition A.11** (Bipartite graph)**.** Repeated from Definition 3.1. A graph $G = (V, E)$ is called *bipartite* if vertices $V$ can be separated in two disjoint subsets $A, B$ with $V = A \cup B$ and $A \cap B = \emptyset$, so that all edges $e \in E$ have form $e = (a, b)$ with $a \in A$ and $b \in B$, i.e. every edge connects a vertex in $A$ to a vertex in $B$ (p. 168 in (Deo, 1974)).

**Definition A.12** (Complete bipartite)**.** Repeated from Definition 3.2. $G$ is called *complete bipartite* if edges between all vertices in $A$ and all vertices in $B$ exist, i.e. $E$ is the Cartesian Product of $A$ and $B$ (p. 192 in (Deo, 1974)).

*Remark* A.13. Any matrix $W$ can be interpreted as the adjacency matrix of an undirected weighted complete bipartite graph. In this case, rows and columns correspond to vertices in $A$ and $B$, respectively. Entries of $W$ correspond to edge weights between all vertices in $A$ and $B$. Given a matrix $W$, we denote the graph induced by $W$ as $G_W = (V_W, E_W)$.

Now, we have gathered all preliminaries to give a proof for Theorem 3.5. For completeness, we repeat Theorem 3.5 with additional explanations:

**Theorem A.14.** *Let $G_W = (V_W, E_W)$ a bipartite weighted graph induced by a matrix $W \in [0;1]^{n \times m}$ as in Definition 3.4. Edge weights of $G_W$ are given by entries of $W$. We assume uniqueness of entries in $W$, which may not be the case in general, but can be achieved by infinitesimal perturbations. Note, that neural persistence (see Definition 3.4) is continuous, therefore infinitesimal perturbations also only have infinitesimal impact on neural persistence. According to Remark A.13, $G_W$ is bipartite and thus we denote the parts as $V_W = A \cup B$ with $A \cap B = \emptyset$.*

*We define*

$$L = \left( \sum_{b \in B} (1 - \max_{a \in A} W_{a,b})^p + \sum_{a \in A} (1 - \max_{b \in B} W_{a,b})^p \right)^{\frac{1}{p}} \tag{16}$$

*and*

$$U = \left( |B \setminus B_{\not\sim A}| + \sum_{b \in B_{\not\sim A}} (1 - \max_{a \in A} W_{a,b})^p + \sum_{a \in A} (1 - \max_{b \in B} W_{a,b})^p \right)^{\frac{1}{p}}, \tag{17}$$

*where*

$$B_{\not\sim A} = \{ b \in B \mid \forall a \in A : b \neq \text{argmax}_{b' \in B} \, W_{a,b'} \}. \tag{18}$$

*$B_{\not\sim A}$ can be thought of as the set of columns whose maximal element does not coincide with the maximal element in any row of $W$.*

*Then, the following inequalities hold:*

$$0 \leq L \leq \text{NP}_p(W) \leq U \leq (n+m)^{\frac{1}{p}}. \tag{19}$$

In the following, we first give a proof for the lower bound, then for the upper bound.

**Lower bound.** We first prove the lower bound $L \leq \text{NP}_p(W)$ ($0 \leq L$ is clear). Let $\text{MST}(G_W) = (V_W, E_{\text{MST}(G_W)})$ be the maximum spanning tree of $G_W$. According to Definition 3.4, neural persistence is calculated from the weights of all edges in $\text{MST}(G_W)$.

First, we equip the maximum spanning tree $\text{MST}(G_W)$ with arborescence structure. According to Definition A.5, we choose an arbitrary vertex $u \in V_W$ as root to make $\text{MST}(G_W)$ a rooted tree. Then, we obtain arborescence structure on $\text{MST}(G_W)$ as in Remark A.7. Note that this only requires an arbitrary choice of the root, but no particular assumptions on the maximum spanning tree or $G_W$ in general. Instead, we are exploiting general properties of spanning trees.

Having introduced arborescence structure, according to Remark A.10 we have a bijection $\varphi_w : V_W \setminus \{u\} \to [0,1]$ from vertices (without the root) to edge weights. Note that in our case, we can assume edge weights to be in range $[0;1]$, because $G_W$ is induced from $W$ and we assume that $W$ has entries bounded below by 0 and above by 1.

Because $\varphi$ and, assuming uniqueness of entries in $W$, thus also $\varphi_w$ is a bijection, it follows that the image of $\varphi_w$ is identical to the set of weights of edges in the maximum spanning tree, i.e.

$$\text{im} \, \varphi_w = \{ \psi(e) \mid e \in E_{\text{MST}(G_W)} \}, \tag{20}$$

where $\psi : E_W \to [0,1]$ is the function that assigns weights to edges of $G_W$, i.e. $\psi$ maps edges to their corresponding weights. As direct consequence, we can rewrite the definition of neural persistence in terms of $\varphi_w$, namely

$$\text{NP}_p(W) = \left( 1 + \sum_{v \in V_W \setminus \{u\}} (1 - \varphi_w(v))^p \right)^{\frac{1}{p}}. \tag{21}$$

Also note that the number of edges in $\text{MST}(G_W)$, which is $|V_w| - 1$ according to Remark A.4, matches the size of $V_W \setminus \{u\}$. For any vertex $v \in V_W \setminus \{u\}$, the value of $\varphi_w(v)$ is bounded from above by the maximum value of any edge connected to $v$. The reason is that $\varphi_w$ is defined as $\varphi_w := \psi \circ \varphi$ and $\varphi$ maps a non-root vertex $v$ to an edge connected to $v$. For example, assume $v \in A$, i.e. $v$ corresponds to a row in $W$. Then,

$$\varphi_w(v) \leq \max_{b \in B} W_{v,b}, \tag{22}$$

because the weights of edges connected to $v$ are given by the entries in the row of $W$ corresponding to $v$. Furthermore, it follows that

$$\left(1 - \max_{b \in B} W_{v,b}\right)^{\frac{1}{p}} \leq (1 - \varphi_w(v))^{\frac{1}{p}}. \tag{23}$$

The same argument holds for vertices in $B$, which correspond to columns of $W$, and thus for all vertices $v \in V_W \setminus \{u\}$. For the root $u$, again by definition of $W$, we get

$$\left(1 - \max_{b \in B} W_{u,b}\right)^{\frac{1}{p}} \leq 1, \tag{24}$$

which corresponds to the constant offset in the definition of neural persistence. Here, we assume $u \in A$, but the same argument can be made when $b \in B$.

In summary, we see that all additive terms in the definition of $L$ are bounded by a corresponding additive term in the definition of $\text{NP}_p(W)$, and so it follows that $L \leq \text{NP}_p(W)$.

**Upper bound.** To prove the upper bound $\text{NP}_p(W) \leq U$, we adopt the filtration perspective detailed by Rieck et al. (2019): We traverse all edges sorted by weight in reversed order, i.e. from largest weight to smallest. Whenever the introduction of an edge connects two previously disconnected components, the edge is saved. The resulting saved edges are the same as the edges that appear in the maximum spanning tree, i.e. $E_{\text{MST}(G_W)}$.

From this perspective we observe that for every vertex $v \in V_W$ the maximum spanning tree necessarily contains the edge connected to $v$ with maximum weight. The reason is that, when traversing edges in descending order sorted by weight, the first edge $e$ connected to any vertex $v$ is the edge connected to $v$ with largest weight. Because we have not encountered another edge connected to $v$ before, $v$ is not connected to any other vertex. Instead, at this point $v$ is an isolated connected component. Thus, by definition of neural persistence, $e$ appears in the calculation of neural persistence.

However, this does not suffice to describe the full maximum spanning tree, because some edges may have maximal weight for both vertices connected by the edge. This is captured by the definition of $U$: The right-most term,

$$\sum_{a \in A} (1 - \max_{b \in B} W_{a,b})^p,$$

captures all edges with maximal weight connected to vertices in $A$. The middle term,

$$\sum_{b \in B_{\not\prec A}} (1 - \max_{a \in A} W_{a,b})^p,$$

captures all edges that have maximal weight for a vertex in $B$ but are not maximal for any vertex in $A$. These weights are necessarily part of the maximum spanning tree.

The last term, $|B \setminus B_{\not\prec A}|$ captures all remaining edges and the constant offset. We do not make any assumptions about them and give them the value that maximises neural persistence, i.e. 0. Since we have $|A| + |B| = |V_W|$ and $|B| = |B_{\not\prec A}| + |B \setminus B_{\not\prec A}|$, the numbers of additive terms in $U$ and the definition of neural persistence match.

Finally, the inequality $U \leq (n + m)^{\frac{1}{p}}$ holds, because here by definition terms of form $(1 - w)^p$ are bounded by 1 and the sum in the definition of $U$ contains exactly $n + m$ terms.

|  |  | | $B$ | | max(rows) |
|---|---|---|---|---|---|
|  |  | $b_1$ | $b_2$ | $b_3$ |  |
| $A$ | $a_1$ | **0.5** | 0.1 | **0.8** | 0.8 |
|  | $a_2$ | **0.7** | **1.0** | 0.1 | 1.0 |
|  | $a_3$ | 0.2 | **0.8** | 0.0 | 0.8 |
| max(cols) |  | 0.7 | 1. | 0.8 |  |

Table 3: Example to illustrate the derived bounds on neural persistence. Bold values are entries of $W$ that appear in the calculation of neural persistence (but not necessarily in the calculation of the bounds). According to max-over-rows values and max-over-columns values, $B_{\not\succ A} = \{b_1\}$ and we have $L \approx 0.45$, $U \approx 1.47$ and $\mathrm{NP}_2(W) \approx 1.29$.

**Example.** We provide an example for the calculation of bounds in Table 3. Note, that this is purely for demonstration purposes, as in practise matrices are much larger.

$L$ is derived by combining the maximum value in each row and the maximum value in each column of $W$. In the example, the multiset (i.e. the set which may contain multiple instances of the same element) of weights used to calculate $L$ is $\{0.8, 1.0, 0.8, 0.8, 1.0, 0.7\}$. $U$ is derived from $L$ by replacing duplicates with 0. Accordingly, the multiset of weights used to calculate $U$ is $\{0.8, 1.0, 0.8, 0.7\}$ and $B \setminus B_{\not\succ A} = \{b_2, b_3\}$, because the weight that is maximal in the column corresponding to $b_2$ is also maximal in the row corresponding to $a_2$, and likewise the weight that is maximal in the column corresponding to $b_3$ is maximal in the row corresponding to $a_1$.

## B   Empirical validation of bounds and implications for neural persistence

### B.1   Tightness of bounds

Here, we evaluate how tight the bounds derived in Theorem 3.5 are using synthetic data. In particular, we sample matrices $W$ with entries sampled iid from truncated Pareto distributions and then control for the spatial concentration of large weights using the method described in Appendix E.1. Sampling from truncated Pareto distributions allows to obtain matrices with neural persistence values that exhaust the full range of possible values, i.e. the interval $[0; 1]$. Furthermore, truncated Pareto distributions are a good model for the distribution of absolute, normalised weights found in trained deep feed-forward neural networks (see Appendix D).

For the resulting matrices, we calculate the respective values of neural persistence, the lower bound $L$ and the upper bound $U$. Additionally, we compute the sortedness sortedness($W$) (see Appendix E.2) for all matrices to show that the upper bound is tighter when sortedness($W$) is larger.

Concretely, we sample 10 000 different matrices with the following parameters:

- Parameter $b$ of truncated Pareto distributions is sampled form a Beta distribution with parameters $\alpha = 1$ and $\beta = 2$, scaled by factor 60. This means that $b = 60x$, $x \sim \text{Beta}(1, 2)$. Sampling values in this way generates matrices whose neural persistence values exhausts the full range of possible values.

- Noise level $s$, where $\log(s)$ is uniformly sampled from the interval $[-10, 0]$. We explain the relationship between noise level $s$ and the resulting sortedness value sortedness($W$) in Appendix E.3. This explanation also motivates our sampling range.

- The number of rows and number of columns sampled uniformly but independently between 50 and 500, i.e. the minimum number of rows (or columns) is 50, and the maximum number of rows (or

| | Absolute error | | Relative error | |
|---|---|---|---|---|
| | Mean | Max. | Mean | Max. |
| Lower bound | 0.004 | 0.043 | 01.04% | 75.44% |
| Upper bound | 0.021 | 0.441 | 13.18% | 1302.40% |

Table 4: Absolute and relative errors of the upper bound and lower bound estimated on 10 000 synthetic matrices. We report mean and maximum deviation from the actual neural persistence value.

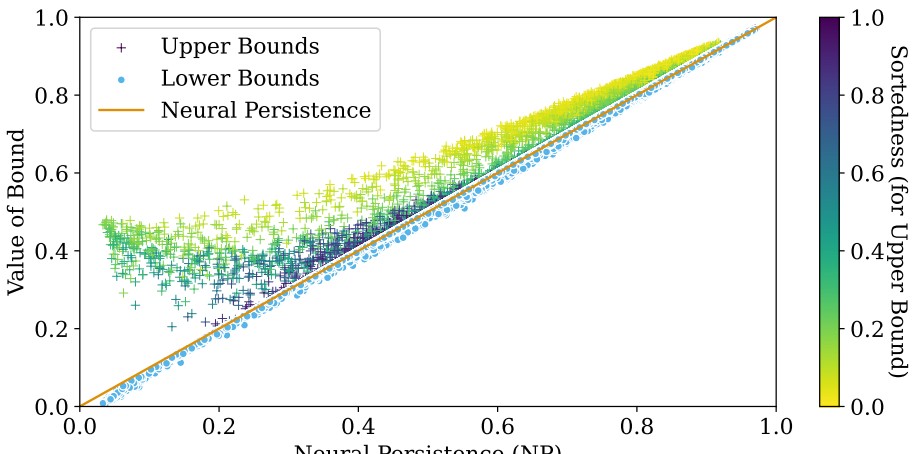

Figure 4: Tightness of bounds on neural persistence in Theorem 3.5. Values above the diagonal correspond to upper bounds, and values below the diagonal correspond to lower bounds. Additionally, we indicate the sortedness of matrices for upper bounds by colour to demonstrate that the upper bound tightens with increasing spatial concentration of large weights. The diagonal is shown as an orange line. Lower bounds are plotted as cyan (blue) circles, and upper bounds are plotted as crosses. A sortedness value of 0.0 means random dispersion of large entries, and a sortedness value of 1.0 means perfect concentration of large entries on bottom rows.

columns) is 500. Matrices are not necessarily square, i.e. the number of rows and the number of columns of a matrix can differ.

Results are in Figure 4. Values above the diagonal correspond to values of the upper bound, and values below the diagonal correspond to values of the lower bound. We plot values of the upper bound in different colours showing the sortedness value sortedness($W$) of the matrix $W$ the upper bound is calculated from. This illustrates that the upper bound is tighter when sortedness is larger. Also, we show the diagonal as an orange line and values corresponding to lower bounds as cyan (blue) circles. To additionally separate values corresponding to upper bounds, we plot them as crosses.

Figure 4 shows that the lower bound is generally tight. Only in the vicinity of 0, i.e. for very small neural persistence values, the tightness of the lower bounds seems to slacken. Contrarily, the upper bound only becomes tight for neural persistence values greater than 0.5. For lower neural persistence values, the bound is only tight when sortedness($W$) is close to 1, i.e. for matrices with high spatial concentration of large weights. Note, however, that in practise for trained models, we mostly encounter neural persistence values greater than 0.5. Therefore, we do not consider the upper bound not being tight when neural persistence is small to be a problem. Furthermore, our insights do not rely directly on the tightness of bounds, but provide theoretical insights into which factors impact neural persistence.

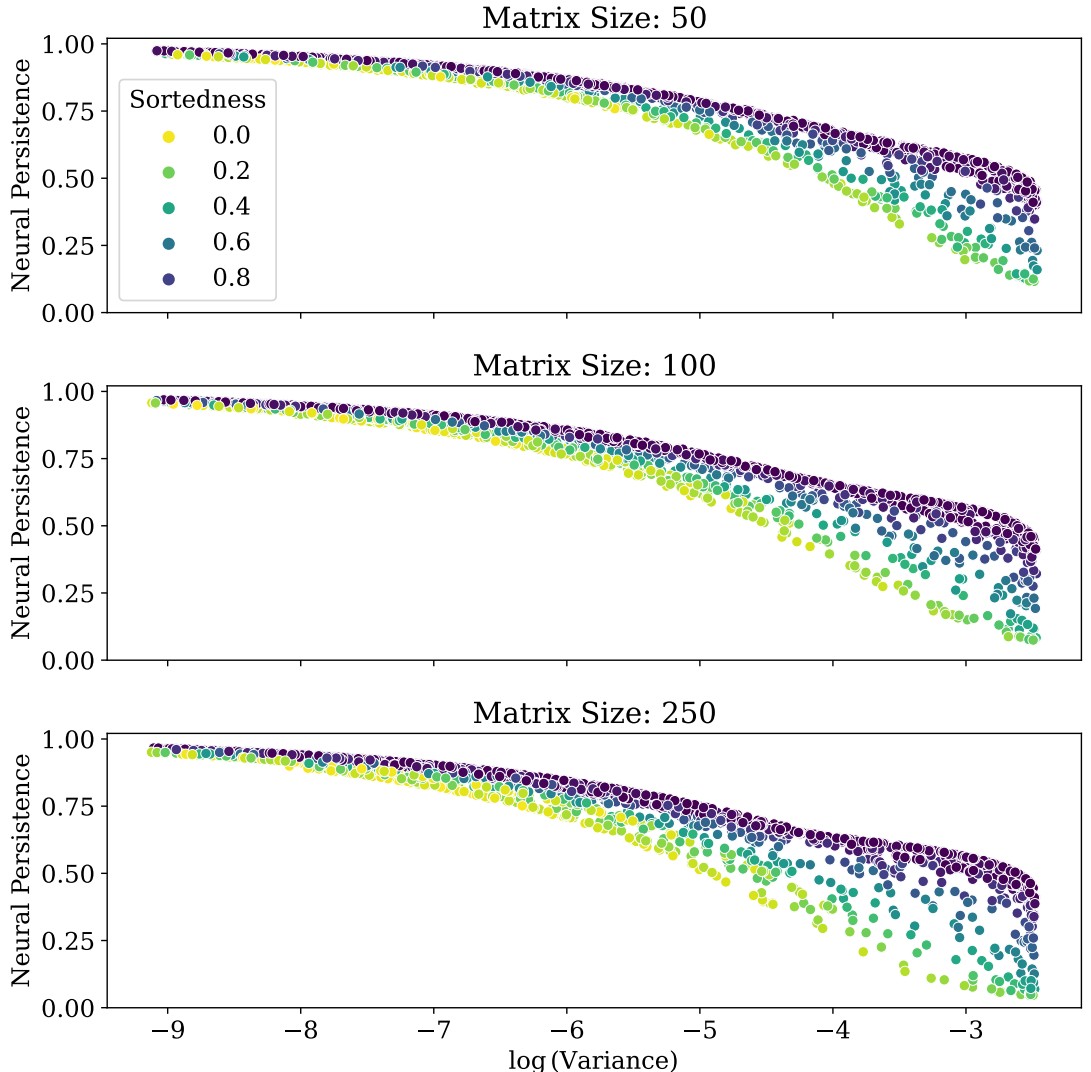

Figure 5: Analysing the neural persistence of synthetic weights matrices when varying the variance and spatial concentration of large weights. We observe approximately linear correspondence of neural persistence with log-variance and spatial concentration of large weights. The $R^2$ score of a linear regression fit is around 89% for both samples from truncated Pareto distributions and samples from truncated normal distributions. "Sortedness" is a proxy measure for the spatial concentration of large weights. A sortedness value of 0 means random spatial dispersion, 1 means that the matrix entries are perfectly sorted.

## B.2 Variance and spatial concentration of large weights as factors that impact neural persistence

We empirically validate the insights obtained from the bounds on neural persistence (3.5). Namely, we claim that variance of weights and spatial concentration of large weights are main factors that impact neural persistence. Also, recall that with weights, we always refer to weights mapped to the range $[0; 1]$ by first taking absolute values and then dividing by the largest absolute weight value.

For our evaluation, we use synthetic, i.e. randomly sampled, weight matrices. For these synthetic matrices, we control the variance of weights and the spatial concentration of large weights. Varying these factors, all else being equal, allows us to investigate their relationship to neural persistence in a controlled way.

We calculate the neural persistence value, the variance, and the sortedness value sortedness($W$) (see Appendix E.2) for all generated matrice $W$s.

In particular, we sample matrices according to the following parameters:

- Entries are sampled iid from truncated Pareto distributions and truncated Normal distributions, so that entries are within the range $[0; 1]$. The type of distribution is sampled randomly for every matrix. Truncated Pareto distributions have one parameter $b > 0$, and truncated normal distributions similarly have a single parameter $\sigma$, which we set to $\frac{1}{b}$ for convenience. We sample $b$ from a Beta distribution with parameters $\alpha = 1$ and $\beta = 2$ scaled by factor 60, i.e. $b = 60x$, $x \sim \text{Beta}(1, 2)$. This gives distributions with different variances to exhaust the full range of neural persistence values.

- The noise level $s$ of the noisy sorting process described in Appendix E.1 that controls the spatial concentration of large weights is sampled as follows: We sample $\log(s)$ uniformly between -10 and 0 and then apply exp to get a value between 0 and 1. The change of sortedness($W$) is faster for sortedness values close to 0 (see Appendix E.3), therefore sampling uniformly on a log-scale gives a better distribution of resulting sortedness values.

- For each number of rows and columns in $\{50, 100, 250\}$, where the number of rows is always equal to the number of columns, we sample 2000 matrices randomly according to the parameters described above. Note, that in this case it is necessary to treat matrices with different sizes separately, because we observe that the neural persistence value also depends on the matrix size (see Appendix C). Showing results for different matrix sizes confirms the generality of our observation and proves that it is not an artifact of a particular set of parameters.

Figure 5 shows that neural persistence increases monotonically with decreasing variance, i.e. lower variance (and in this case also mean weight value) results in larger neural persistence. The increase of neural persistence with decreasing variance is approximately linear when log-transforming the variance. Likewise, for fixed variance, neural persistence increases monotonically with sortedness. Fitting a linear regression model which maps the log-variance and sortedness to neural persistence achieves a coefficient of determination $R^2 \approx 89\%$ for all matrix sizes. The $R^2$ score measures the proportion of variance in neural persistence that is predictable from log-variance and sortedness. Therefore, this result means that a significant amount of the variance in neural persistence can be explained from variance of weights and sortedness.

In summary, these observations confirm a strong relationship between the variance and spatial concentration of weights and neural persistence.

## C   Normalised neural persistence depends on matrix size

Here, we provide empirical evidence that the normalisation of neural persistence to the range $[0, 1]$ as proposed by Rieck et al. (2019) does not achieve comparability of values calculated for different matrix shapes. Recall that the neural persistence of a $n \times m$ matrix with entries in $[0; 1]$ is bounded by $(n + m - 1)^{\frac{1}{p}}$. This is asserted by Theorem 1 in (Rieck et al., 2019). This bound is used to normalise neural persistence values to the interval $[0; 1]$, with the aim of comparing neural persistence values calculated for matrices of different sizes. However, we find that even this definition of normalised neural persistence tends to decrease with increasing matrix size. In other words, using the theoretical upper bound for normalisation does not lead to comparability between matrix sizes.

However, this is not a theoretical issue: The constructive proof of the bounds on neural persistence (Theorem 1 in (Rieck et al., 2019)) shows that, for any matrix size, a matrix can be constructed whose neural persistence is equal to the upper bound. The theoretical upper bound is a limiting case of ever more positive-skewed distributions of normalised absolute weights. It is reached when all edges except for one have weight 0, and the remaining edge has weight 1. Therefore, maximum neural persistence is approached by having increasingly skewed distributions of weights. However, when fixing a certain distribution of weights, increasing the matrix size leads to a decrease of neural persistence.

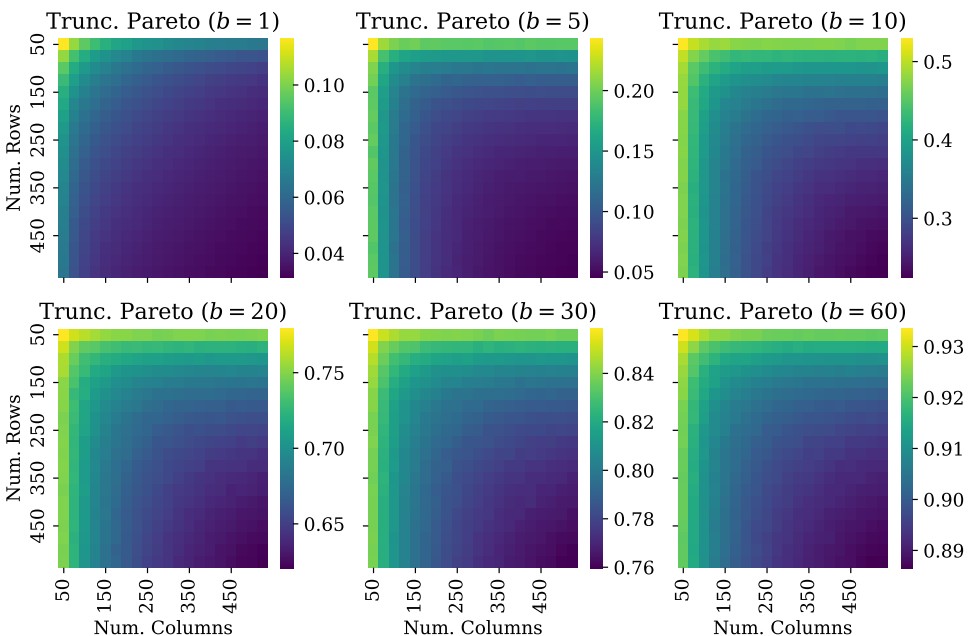

Figure 6: Expected normalised neural persistence for different matrix sizes and Truncated Pareto distributions with varying parameter $b$. The expected normalised neural persistence decreases with increasing matrix size (darker is lower). Note that heatmaps have different scales. Our aim is to show the trend, not to compare values for different distributions.

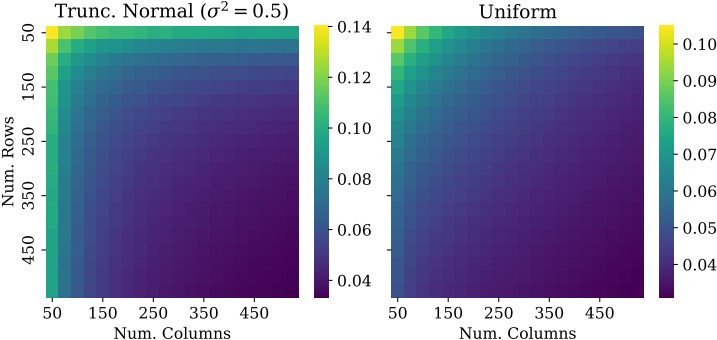

Figure 7: Expected normalised neural persistence for different matrix sizes and a Truncated Normal distributions with parameter $\sigma^2 = \frac{1}{2}$ as well as a Uniform distribution. The expected normalised neural persistence decreases with increasing matrix size (darker is lower). Note that heatmaps have different scales. Our aim is to show the trend, not to compare values for different distributions.

We demonstrate this in the following way: For given $n, m$ which specify the matrix size and distribution $P$ defined on the unit interval, we sample matrices of size $n \times m$ with entries sampled iid from $P$. Then, we compute the mean normalised neural persistence from all sampled matrices.

In particular, we demonstrate the trend by sampling weights from a Uniform Distribution, a Truncated Normal Distribution with $\sigma = 0.5$, and Truncated Pareto Distributions with increasing parameter $b$. The Uniform Distribution and the extremely skewed Truncated Pareto Distribution with parameter $b = 60$ represent different extrema that may be encountered. Therefore, showing that the claimed trend holds for both distributions affirms its generality.

Results for Truncated Pareto distributions are in Figure 6. Results for the Truncated Normal distribution and the Uniform distribution are in Figure 7. The trend that normalised neural persistence decreases with increasing matrix size, i.e. larger number of rows or columns, is visible for all distributions. Note, that concrete neural persistence values differ for each distribution, but here we are only interested in the trend.

For a theoretical justification, consider the argument used to derive bounds in Theorem 3.5 (see Appendix A): There is a bijection between vertices (excluding one arbitrary root) and the edges in the maximum spanning tree that are used for calculating neural persistence. In other words, we can construct the maximum spanning tree by picking for each vertex exactly one edge connected to the vertex. Assume now that the rank of all selected edges, when sorting edges connected to a vertex by weight in decreasing order, is bounded by $k$. $k$ could be constant or growing very slowly with matrix size. This is a reasonable assumption, because for most vertices, the selected edge has rank 1. Assume further that edge weights are sampled iid from some distribution defined on the unit interval. Then, for any $\epsilon > 0$, the probability of any value larger than $1 - \epsilon$ being withing the top $k$ largest values increases monotonically with matrix size. The reason is that, due to iid sampling, values in each row and column constitute themselves an iid sample. By increasing the matrix size, we increase the sample size and thereby the probability of sampling a value close to the maximum, i.e. 1. Note that larger weight values lead to lower neural persistence.

## D    Distribution of weights in trained neural networks

Here, we quantify the distribution of weights actually encountered in the models trained as described in Section 4.1. Our concrete aim is to make our experiments in Appendix B as faithful to real data as possible. The main finding in this section is that normalised absolute weight values in trained neural networks are best modelled by truncated Pareto distributions and truncated Normal distributions. In consequence, we use these two families of distributions whenever we sample synthetic matrices.

**Method** We propose the following method for analysing weight distributions: Given a weight matrix $W \in [0;1]^{n \times m}$, we fit parameters of distributions defined on the unit interval by minimising the negative log-likelihood of weights in $W$. This means that for a given family of univariate distributions with parameters $\theta$, we minimize

$$\mathcal{L} = -\log \prod_{i,j} P(W_{i,j} \mid \theta) = -\sum_{i,j} \log P(W_{i,j} \mid \theta). \tag{25}$$

Weight matrices $W$ are obtained from the weight matrices in trained neural networks by first taking absolute values of all entries and then dividing each entry by the globally maximal absolute weight value in the entire neural network, i.e. applying the same normalisation that is applied to calculate neural persistence values.

Since we use various distributions where no closed-form solution for $\mathrm{argmin}_\theta \mathcal{L}$ exists, we use a black-box optimiser to find optimal parameters. Concretely, we use the Nelder-Mead method (Nelder & Mead, 1965) implemented in `scipy.optimize.minimize`. Using a black-box optimizer is feasible, because we only consider families of distributions with at most two parameters, which means that fitting parameters is computationally relatively cheap. Also, in our case all parameters are constrained to positive values. We realise this constraint by optimizing the log-transformed value of the parameters which is unconstrained in $\mathbb{R}$, and apply exp after optimisation.

Anecdotally, it is well known that trained neural networks tend to have very skewed distributions of weight magnitudes with few large weights and most weights being close to 0. Therefore, expecting skewed distributions of weights, we consider the following families of distributions and fit their parameters for every weight matrix of all trained models:

- Beta distribution with parameters $\alpha > 0$ and $\beta > 0$

- Truncated Exponential distribution with parameter $\lambda > 0$

- Truncated Normal distribution with parameter $\sigma > 0$ (location is fixed to 0)

- Truncated Pareto distribution with parameter $b > 0$

Note, that all truncated distributions are truncated in a way to be defined on the unit interval. For Beta distributions, this is the case by definition. Therefore, our analysis is limited by the four families of distributions that we consider. However, we the Beta distributions is very flexible and therefore a strong baseline when comparing to the three truncated distributions, which are biased towards modelling skewed distributions.

**Results.** First, we analyse which distributions are best models of weights in trained neural networks. For each weight matrix, we compare the negative log-likelihood $\mathcal{L}$ of entries achieved by the optimised parameters. We assume that the distribution whose optimal fit to the weights yields the lowest negative log-likelihood is the best model of the entries found in the respective matrix. Note, that we do not take the numbers of parameters into account and only compare log-likelihoods. Also, it may be the case that for some matrices the optimiser fails or does not find optimal values. However, given that we analyse around 5000 matrices in total, this does not obscure the (clear) trends visible in our results.

In Figure 8, we show the relative win rates of each distribution family for models trained on different datasets and with different numbers of layers. We also show results for each layer separately, but aggregate over hidden sizes and different activation functions. We define the win rate as the ratio of considered matrices where the respective family of distributions achieves lowest negative log-likelihood. This means that the win rates sum to one for the four considered families of distributions.

The general trend is that weights of deeper models and in later layers of all models are better modelled by truncated Pareto distributions. However, especially for models trained on EMNIST and Fashion-MNIST, weights in the first layer, i.e. the layer connected to inputs, are best modelled by truncated normal distributions. The relatively high number of winning truncated normal distributions in the final layer of one layer models is mostly due to models with tanh activation function. Perhaps surprisingly, Beta distributions rarely yield best negative log-likelihood, which confirms our initial hypothesis about distributions of weight magnitudes being very skewed.

Next, we are interested in the values of fitted parameters of truncated Pareto distributions and truncated normal distributions. Through this, we can assess how skewed distributions actually are and if there are any perceivable trends. Figure 9 shows the distributions of optimal values for the parameter $b$ of truncated Pareto distributions. We distinguish between models with different numbers of layers trained on different datasets, and show results for each layer individually. However, we aggregate over models with different hidden sizes and activation functions. In all cases, the main trend is that later layers have a less skewed distribution of weight magnitudes (lower value of $b$). We hypothesise that in the first layer, some features receive very high weights (and in fact it can be shown that concentration of large weights on fewer features monotonically increases during training), whereas later layers are more densely and also more evenly connected. The observed trend is confirmed when looking at optimal parameters of truncated normal distributions (see Figure 10). Here, values of parameter $\sigma$ are higher in later layers, indicating less skewed distributions.

## E   Manipulating the spatial concentration of large entries in matrices

### E.1   Noisy sorting process

Here, we explain how to obtain matrices with controlled spatial concentration of large weights. The core of the method is to reorder entries of a matrix $W$ so that they correspond to a non-perfect sorting of the entries. This is achieved by constructing a permutation $\pi$ of entries that sorts a perturbed matrix $W_{\text{noise}}$. Then, we apply the permutation $\pi$ to $W$ to reorder entries. We denote the result of applying the permutation $\pi$ to $W$ as $W_{\text{sort}}$.

$W_{\text{noise}}$ is given by $W + s \cdot \epsilon$, where $s \in \mathbb{R}_+$ is a scalar parameter that determines the noise level. $\epsilon$ has the same dimension, i.e. number of rows and columns, as $W$. Each entry $\epsilon_{i,j}$ of $\epsilon$ is sampled iid from a standard normal distribution, i.e. $\epsilon_{i,j} \sim \mathcal{N}(0,1) \; \forall i,j$. Due to reparametrization, this means that $s$ is equal to the standard deviation of the added noise. Also note, that $W_{\text{noise}}$ has the same dimension, i.e. number of rows and columns, as $W$.

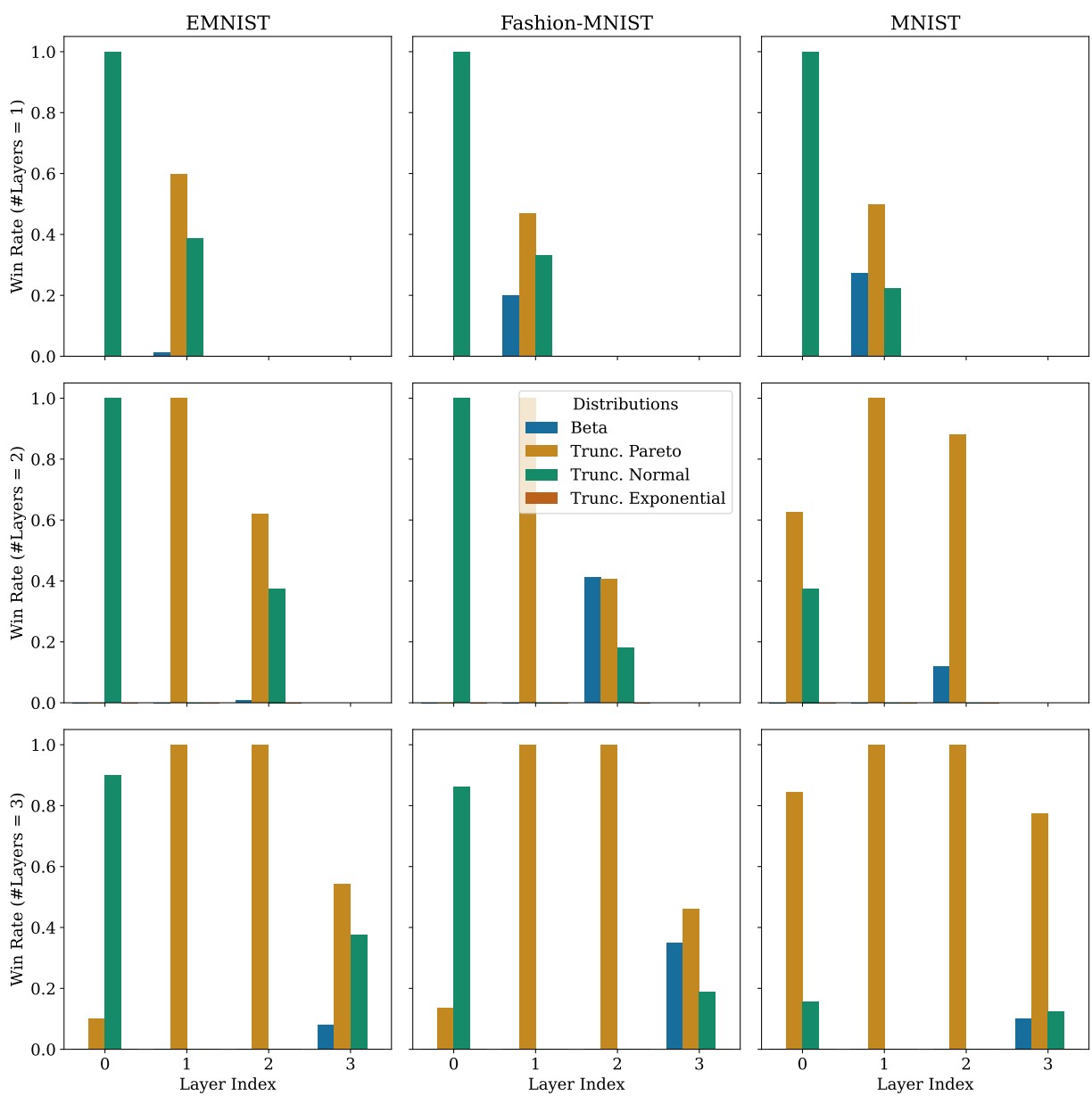

Figure 8: Ratios of weight matrices where different families of distributions achieve highest log-likelihood of normalised entries after fitting parameters. We show results for different datasets and models factorised by number of layers in the model and weight matrices factorized by position in the model.

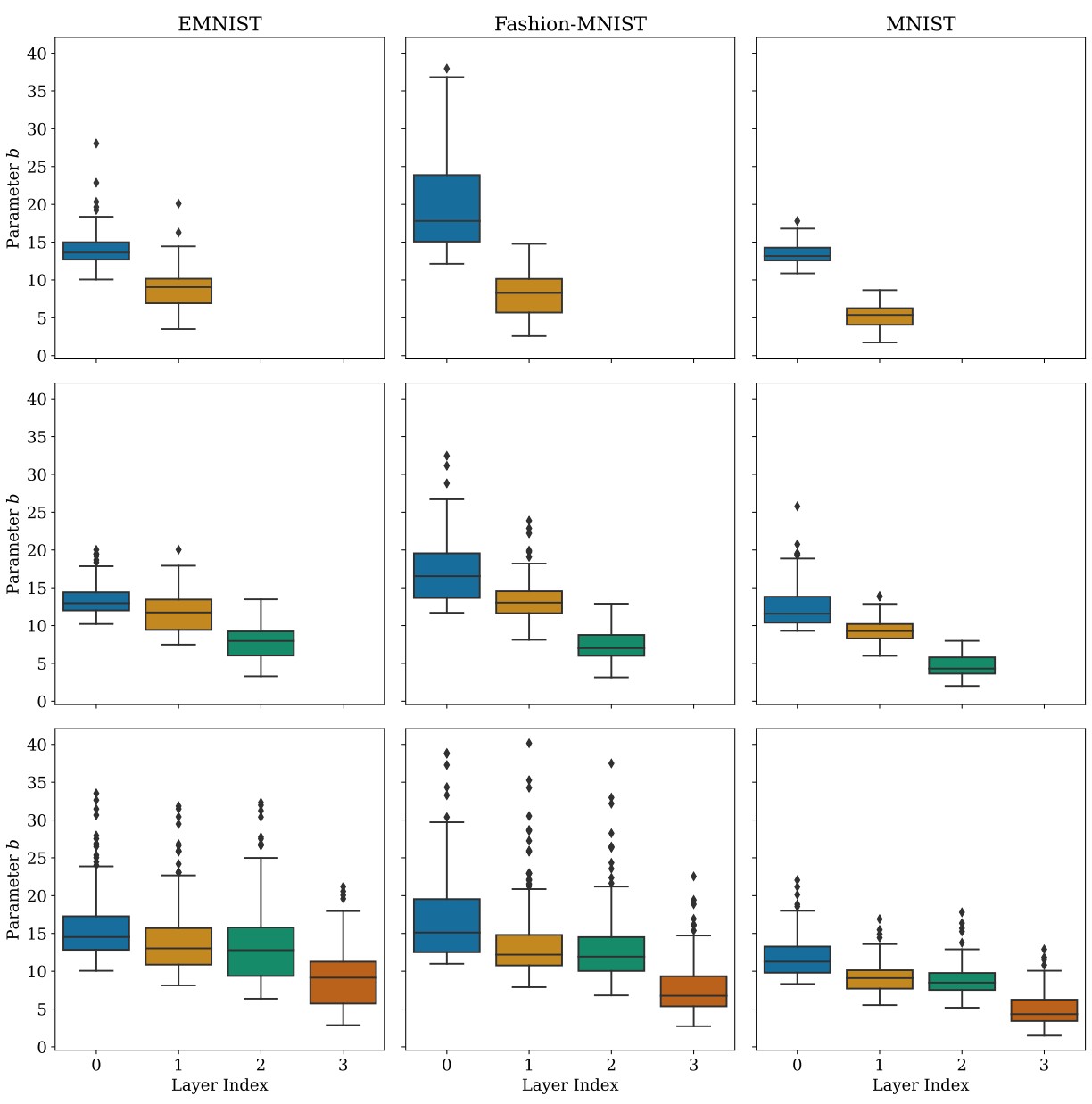

Figure 9: Optimal values of parameter $b$ for truncated Pareto distributions shown for models with different number of layers trained on different datasets. We show values for each layer separately, but aggregate values for models with different hidden sizes and activation functions. We observe that later layers generally have lower value of $b$, which indicates less skewed distributions.

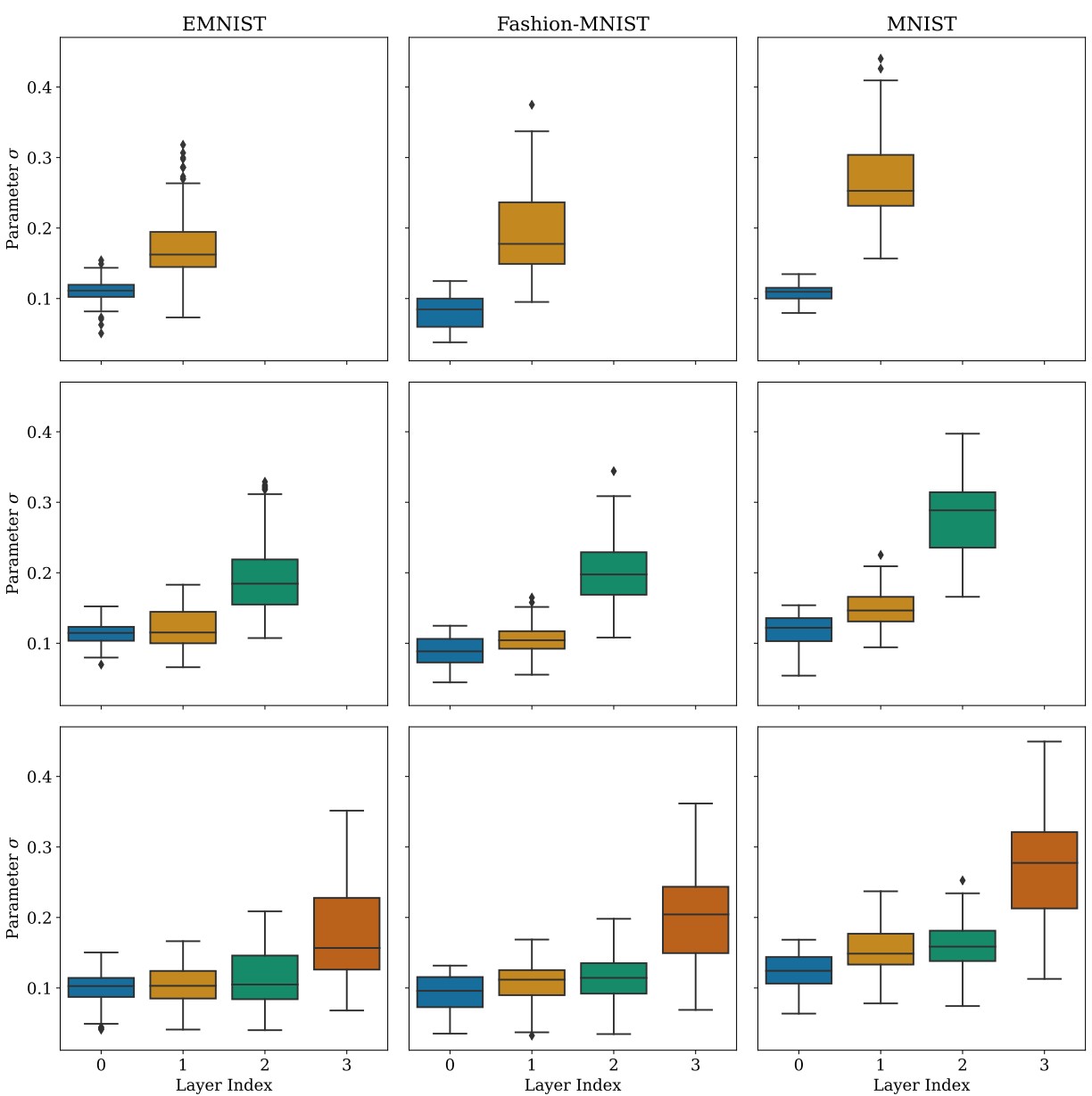

Figure 10: Optimal values of parameter $\sigma$ for truncated normal distributions shown for models with different number of layers trained on different datasets. We show values for each layer separately, but aggregate values for models with different hidden sizes and activation functions. We observe that later layers generally have higher value of $\sigma$, which indicates less skewed distributions.

Next, we formally describe how to derive $\pi$. First, we define two total orders on matrices $W$. We denote the index set of $W$, i.e. the set of all (row, column) index pairs that specify positions in $W$, as $\mathcal{I}$. Note that $\mathcal{I}$ is also the index set of $W_{\text{noise}}$.

**Definition E.1** (Total order on matrix entries from indices). Let $I = (i, j) \in \mathcal{I}$ and $I' = (i', j') \in \mathcal{I}$ two pairs of (row, column) indices that specify positions of entries in $W$. Then, define the relation $\leq_{\text{index}}$ as

$$I \leq_{\text{index}} I' \iff (i \leq i') \vee (i > i' \wedge j \leq j'). \tag{26}$$

In other words, an index pair $I = (i, j)$ precedes another index pair $I' = (i', j')$ if $I$ indexes an entry of $W$ that has lower row index as $I'$ or the same row index but lower column index. Clearly, this is a total order on the index set of $W$.

**Definition E.2** (Total order on matrix entries from values). Let $I = (i, j) \in \mathcal{I}$ and $I' = (i', j') \in \mathcal{I}$ two pairs of (row, column) indices that specify positions of entries in $W$. Then, define the relation $\leq_{\text{value}}$ as

$$I \leq_{\text{value}} I' \iff W_{\text{noise}}[i, j] \leq W_{\text{noise}}[i', j']. \tag{27}$$

In other words, an index pair $I = (i, j)$ precedes another index pair $I' = (i', j')$ if $I$ indexes an entry of $W$ that has lower value as the entry indexed by $I'$.

If we assume uniqueness of values of entries of $W$ and $W_{\text{noise}}$, which in general is not the case but can be achieved by infinitesimal perturbations, both $\leq_{\text{index}}$ and $\leq_{\text{value}}$ are strict total orders. Then, it is a well know fact that finite totally ordered sets of size $k$ can be indexed by the first $k$ natural numbers. In other words, for both $\leq_{\text{index}}$ and $\leq_{\text{value}}$, a bijection $\iota : \mathcal{I} \to \mathbb{N}_{\leq |\mathcal{I}|}$ exists that maps $\mathcal{I}$ (ordered according to the respective relation) to the first $k$ natural numbers. We denote the bijection induced by $\leq_{\text{index}}$ as $\iota_{\text{index}}$, and the bijection induced by $\leq_{\text{value}}$ as $\iota_{\text{value}}$.

Now, we define $\pi$ as a bijection

$$\begin{aligned} \pi &: \mathcal{I} \to \mathcal{I}, \\ (i, j) &\mapsto (\iota_{\text{value}}^{-1} \circ \iota_{\text{index}})((i, j)). \end{aligned} \tag{28}$$

Then, applying $\pi$ to $W$, i.e. changing entries of $W$ according to the permutation $\pi$, yields an approximately sorted matrix $W_{\text{sort}}$. $W_{\text{sort}}$ is only approximately sorted, because the order induced by $\leq_{\text{value}}$ when using values in $W_{\text{noise}}$ does not fully coincide with the order induced by $\leq_{\text{value}}$ when using values in $W$ for sorting due to the added Gaussian noise.

From this consideration, it is clear that the scalar noise parameter $s$ controls the deviation of $W_{\text{noise}}$ from a perfectly sorted matrix. Depending on the noise parameter $s$, large values of $W$ concentrate towards the bottom rows of $W_{\text{sort}}$. In the case of $s = 0$, $W_{\text{sort}}$ is perfectly sorted. For large $s$, $W_{\text{sort}}$ is more like a random permutation of $W$.

### E.2 Sortedness criterion

Here, we formally describe our sortedness criterion that we use to measure the spatial concentration of large weights in any given matrix $W$. We denote the criterion as sortedness$(W)$. The criterion measures the number of inversions needed to sort the matrix $W$. This was proposed as a useful measure of the sortedness of arrays in (Mannila, 1985). As underlying order on matrix entries we again use $\leq_{\text{index}}$ (see Definition E.1) and $\leq_{\text{value}}$ (see Definition E.2) from Appendix E.1. However, as values used for comparison in the definition of $\leq_{\text{value}}$ we use entries in $W$, i.e. the argument of sortedness, not $W_{\text{noise}}$.

The number of inversions is formally defined as the cardinality $|\text{Inv}|$ of the set $\text{Inv} \subset \mathcal{I} \times \mathcal{I}$ that is defined as

$$\text{Inv} = \{((i, j), (i', j')) \in \mathcal{I} \times \mathcal{I} \mid (i, j) \leq_{\text{index}} (i', j') \wedge (i', j') \leq_{\text{value}} (i, j)\} \tag{29}$$

This means that all tuples of index pairs $((i, j), (i', j'))$ are included in Inv where the order induced by indices does not correspond to the order induced by values.

To project score values into the range $[0; 1]$, we normalise by the maximum size of Inv. The maximum size of Inv is $\frac{|\mathcal{I}| \cdot (|\mathcal{I}| - 1)}{2}$ Also, we substract the normalised score from 1 in order to get a measure of ascending

sortedness, not descending sortedness. Based on these considerations, we define a metric $\Omega$ which measures the spatial concentration of large entries on bottom rows in matrices as

$$\Omega(W) \overset{\text{def}}{=} 1 - \frac{2 \cdot |\text{Inv}|}{|\mathcal{I}| \cdot (|\mathcal{I}| - 1)} \tag{30}$$

According to Estivill-Castro (2004), $\Omega$ is closely related to the Kendall rank correlation coefficient (Kendall, 1938), otherwise known as Kendall's $\tau$. First, we define the complement of Inv, namely

$$\text{Ninv} \overset{\text{def}}{=} \{((i, j), (i', j')) \in \mathcal{I} \times \mathcal{I} \mid (i, j) \leq_{\text{index}} (i', j') \wedge (i, j) \leq_{\text{value}} (i', j')\} \tag{31}$$

with the property, that

$$\text{Inv} \cup \text{Ninv} = \{((i, j), (i', j')) \in \mathcal{I} \times \mathcal{I} \mid (i, j) \leq_{\text{index}} (i', j')\} \tag{32}$$

and thus

$$|\text{Ninv}| = |\mathcal{I}| \cdot (|\mathcal{I}| - 1) - |\text{Inv}|. \tag{33}$$

These definitions allow to redefine $\Omega$ as

$$\Omega(W) = 1 - \frac{2 \cdot |\text{Inv}|}{|\mathcal{I}| \cdot (|\mathcal{I}| - 1)} = \frac{|\mathcal{I}| \cdot (|\mathcal{I}| - 1)}{|\mathcal{I}| \cdot (|\mathcal{I}| - 1)} - \frac{2 \cdot |\text{Inv}|}{|\mathcal{I}| \cdot (|\mathcal{I}| - 1)} = \frac{2 \cdot |\text{Ninv}|}{|\mathcal{I}| \cdot (|\mathcal{I}| - 1)} \tag{34}$$

Thus, transforming $\Omega$, we have

$$2\Omega(W) - 1 = \frac{4|\text{Ninv}|}{|\mathcal{I}| \cdot (|\mathcal{I}| - 1)} - 1 = \frac{4|\text{Ninv}|}{|\mathcal{I}| \cdot (|\mathcal{I}| - 1)} - \frac{2 \cdot (|\text{Ninv}| + |\text{Inv}|)}{|\mathcal{I}| \cdot (|\mathcal{I}| - 1)} = \frac{2}{|\mathcal{I}| \cdot (|\mathcal{I}| - 1)} \cdot (|\text{Ninv}| - |\text{Inv}|) \tag{35}$$

which corresponds exactly to the definition of the Kendall rank correlation coefficient when comparing to an array perfectly sorted in ascending order. Since efficient implementations for calculating the Kendall rank correlation coefficient exist, we define our sortedness measure sortedness($W$) as

$$\text{sortedness}(W) \overset{\text{def}}{=} 2\Omega(W) - 1 \tag{36}$$

and use the SciPy implementation `scipy.stats.kendalltau` to compute sortedness($W$).

In practise, an (approximate) correspondence of $\leq_{\text{index}}$ and $\leq_{\text{value}}$ cannot be expected in arbitrary matrices, even if they exhibit spatial concentration of large weights. In particular, matrices may exhibit spatial concentration of large entries in certain rows, certain columns, or both. Therefore, when calculating sortedness for arbitrary matrices, we first permute rows and columns, so that they are sorted in ascending order by mean value (over rows or columns, respectively). Furthermore, we compute sortedness using $\leq_{\text{index}}$ for $W$, its transpose $W^T$ and also using an order on matrix entries $\leq_{\text{diagonal}}$ in place of $\leq_{\text{index}}$, that is induced by diagonals as follows:

**Definition E.3** (Total order on matrix entries from diagonals). Let $I = (i, j) \in \mathcal{I}$ and $I' = (i', j') \in \mathcal{I}$ two pairs of (row, column) indices that specify positions of entries in $W$. Then, define the relation $\leq_{\text{diagonal}}$ as

$$I \leq_{\text{index}} I' \iff (i + j) < i' + j' \vee i + j = i' + j' \wedge i < i'). \tag{37}$$

In other words, an index pair $I = (i, j)$ precedes another index pair $I' = (i', j')$ if $I$ indexes an entry of $W$ that lies on a diagonal (from top right to bottom left) that is more to the left or more to the top of the entry in $W$ specified by $I'$. Clearly, this is a total order on the index set of $W$.

This definition accounts for cases where large entries concentrate primarily on an intersection of some columns and rows. Finally, we use the maximum of these three values as the actual sortedness value.

### E.3 Relationship between the noisy sorting process and the sortedness criterion

In this section, we analyse the relationship of the noisy sorting process described in Appendix E.1 and the sortedness criterion described in Appendix E.2. On the one hand, we use the noisy sorting process to generate synthetic matrices with controlled spatial concentration of large weights on the bottom rows. On the other hand, we use the sortedness criterion, denoted as sortedness($W$), to measure spatial concentration of large weights in matrices.

First, we note that the sortedness($W$) is an empirical measure that can be applied to any matrix $W$. Contrarily, the noise level $s$ that parametrises the noisy sorting process cannot be recovered without knowing the original matrix before the sorting process. Therefore, the noisy sorting process does not directly yield an empirical criterion to assess the spatial concentration of large weights in arbitrary matrices. This answers the question why we do not simply use the noise level $s$ that parametrises the noisy sorting process as measure of spatial concentration in the empirical validation part of Section 3.

In order to understand the relationship between the noisy sorting process and the sortedness measure developed in Appendix E.2, we make the following observation about $\Omega$ defined in (34): $\Omega(W)$ can be seen as a (normalised) sum of indicator variables, namely whether any index pair $(i, j) \in \mathcal{I} \times \mathcal{I}$, $i < j$, is not in the set Inv. Therefore, by linearity of expectation, the expected value of $1 - \Omega(W)$ is equal to the probability of any such index pair $(i, j)$ being an element of Inv. In consequence, we have to describe this probability.

For this analysis, we assume that entries of $W$ are sampled iid from some probability distribution $P$ defined on the unit interval, i.e. with support on $[0; 1]$. We define the random variable $\delta$ as the absolute difference between two values sampled from $P$. For example, the expected value of $\delta$ is defined as

$$\mathbb{E}[\delta] = \mathbb{E}_{x,y \sim P}\left[|x - y|\right] \leq 1. \tag{38}$$

Remember from Appendix E.1 that $W_{\text{noise}}$ is obtained from $W$ by adding element-wise Gaussian noise with mean 0 and variance $s^2$ to $W$. For any index pair $((i, j), (i', j')) \in \mathcal{I} \times \mathcal{I}$, let $\epsilon_{i,j}, \epsilon_{i',j'} \sim \mathcal{N}(0, s)$ be the respective random noise values. For simplicity, we denote $I = (i, j)$ and $I' = (i', j')$. Note, that in this scenario, we assume $I \leq_{\text{index}} < I'$ and $W_{i,j} < W_{i',j'}$. Then, with high probability, we have

$$1 - \mathbb{E}[\Omega(W)] = \Pr[(I, J) \in \text{Inv}] = \Pr[W_{i,j} + \epsilon_{i,j} > W_{i',j'} + \epsilon_{i',j'}] = \Pr[(\epsilon_{i,j} - \epsilon_{i',j'}) - (W_{i',j'} - W_{i,j}) > 0]. \tag{39}$$

Unfortunately, we have no prior knowledge about the distribution of $\delta$ and therefore can not evaluate this probability. One practical option is to make the simplifying assumptions that $\delta$ is normally distributed with mean $\mu_\delta$ and variance $\sigma_\delta^2$, which we can estimate from the entries in $W$. In this case, the quantity $(\epsilon_{i,j} - \epsilon_{i',j'}) - (W_{i',j'} - W_{i,j})$ is a sum of three Gaussian random variables, and is accordingly distributed as $\mathcal{N}(-\mu_\delta, 2s^2 + \sigma_\delta^2)$. Now, we can evaluate $\Pr[(\epsilon_{i,j} - \epsilon_{i',j'}) - (W_{i',j'} - W_{i,j}) > 0]$ as

$$\Pr[(\epsilon_{i,j} - \epsilon_{i',j'}) - (W_{i',j'} - W_{i,j}) > 0] = \Phi_{-\mu_\delta, 2s^2 + \sigma_\delta^2}(0) = 1 - \Phi_{0, 2s^2 + \sigma_\delta^2}(\mu_\delta),$$

where $\Phi_{\mu, \sigma^2}$ is the cumulative distribution function of a normal distribution with mean $\mu$ and variance $\sigma^2$. Recalling the definition of the cumulative distribution function of normal distributions, namely

$$\Phi_{0, 2s^2 + \sigma_\delta^2}(\mu_\delta) = \frac{1}{2}\left(1 + \text{erf}\left(\frac{\mu_\delta}{\sqrt{2} \cdot \sqrt{2s^2 + \sigma_\delta^2}}\right)\right),$$

we get the following relationship:

$$\mathbb{E}[\text{sortedness}(W)] = \text{erf}\left(\frac{\mu_\delta}{\sqrt{2} \cdot \sqrt{2s^2 + \sigma_\delta^2}}\right)$$

which we can solve for $s$ when we want to generate matrices with a specific sortedness value. Note, that in reality $\delta$ will generally not be distributed according to a normal distribution. Therefore, this way of finding suitable values of $s$ will be biased and possibly inaccurate.

# F    Algorithm for calculating the summary matrix for deep graph persistence

In Algorithm 1, we show pseudocode for calculating the summary matrix $S$ needed for deep graph persistence (see Definition 5.1). Note, that the syntax is inspired by `numpy` code. This includes the broadcasting semantics for calculating element-wise minima of combinations of entries.

The intuition of Algorithm 1 is as follows: We iterate through weight matrices from output units towards the input units. At the end of each iteration, $S$ holds the minimum weight on the maximum path from each input (to the current layer) to each output unit of the network. Entries of $S$ are updated by first calculating whether the weight from an input is smaller than the path from its target unit to the respective output unit. Then, for each combination of input unit (of the current layer) and output unit, the maximum path is the maximum path from the input unit to the output unit via any target unit in the current layer.

**Data:** ordered list of weight matrices $W$
**Result:** summary matrix $S$
$W \leftarrow \text{reverse}(W)$ ;                              /* Go from output units towards inputs */
$S \leftarrow \text{pop}(W, 0)$ ;                    /* Remove first weight matrix and save in $S$ */
**for** $w$ **in** $W$ **do**
   $S_{\text{expanded}} \leftarrow \text{expand\_dims}(S, 1)$ ;            /* Insert axis for broadcasting semantics */
   $w_{\text{expanded}} \leftarrow \text{expand\_dims}(w, 2)$ ;            /* Insert axis for broadcasting semantics */
   $S \leftarrow \text{elementwise\_minimum}(S_{\text{expanded}}, w_{\text{expanded}})$ ;         /* Minimum of all combinations of
   weights in matrix and paths to output units */
   $S \leftarrow \text{max\_along\_axis}(S, 2)$ ;  /* Maximum path between each input unit and output unit */
**end**
**return** $S$

**Algorithm 1:** Algorithm for calculating the summary matrix $S$ for deep graph persistence

# G    Technical details of image corruptions

Here, we describe in detail how we apply corruptions to images in order to create synthetic data for the covariate shift detection evaluation in Section 5.2. We evaluate four types of corruption, namely Gaussian blur, Gaussian noise, pixel dropout, and uniform noise. In the case of Gaussian noise and uniform noise, feature values are clipped between 0 and 1, to avoid illegal values, which are trivial to detect as corrupted. For each type of corruption, we evaluate six levels of corruption intensity. For Gaussian noise and uniform noise, the corruption intensity is defined by multiplying standard normal or uniform noise with factor $\sigma$. For Gaussian blur, the corruption intensity is defined by the standard deviation $\sigma$ and the kernel size of blurring. For pixel dropout, the corruption intensity is defined by the dropout probability $p$.

To set parameters of corruptions, we use the same parameters as Hensel et al. (2023) for Gaussian blur and Gaussian noise. Additionally, we use the same multipliers for Gaussian noise and for uniform noise. For pixel dropout, we use custom values. An overview over parameters is given in Table 5.

Additionally, we create samples with varying ratio $\delta$ of corrupted samples, while all other images in the sample are non-corrupted. We include corrupted image ratios of $\delta \in \{0.25, 0.5, 0.75\}$. In samples with less corrupted images, it is less likely that the KS test will reject the null hypothesis, therefore such samples are harder to detect by the method evaluated in this work.

# H    Additional results

## H.1    Correspondence of variance and neural persistence for deep neural networks with different hyperparameters

Here, we demonstrate the approximately linear correspondence of neural persistence and log-transformed variance of normalised weights for all deep neural networks (see Section 4.1). For each combination of hyperparameters, we train 20 models with different initialisation and minibatch trajectories. This leads to

| Corruption | | Intensity Level | | | | | |
|---|---|---|---|---|---|---|---|
| | | I | II | III | IV | V | VI |
| | | MNIST | | | | | |
| Gaussian blur | $\sigma$ | 0.35 | 0.4 | 0.5 | 0.6 | 0.7 | 0.8 |
| | $\kappa$ | (3, 3) | (3, 3) | (3, 3) | (3, 3) | (3, 3) | (3, 3) |
| Gaussian/uniform noise | $\sigma$ | $\frac{25}{255}$ | $\frac{40}{255}$ | $\frac{55}{255}$ | $\frac{70}{255}$ | $\frac{85}{255}$ | $\frac{100}{255}$ |
| Pixel dropout | $p$ | 0.1 | 0.2 | 0.3 | 0.4 | 0.5 | 0.6 |
| | | Fashion-MNIST | | | | | |
| Gaussian blur | $\sigma$ | 0.35 | 0.4 | 0.5 | 0.6 | 0.7 | 0.8 |
| | $\kappa$ | (3, 3) | (3, 3) | (3, 3) | (3, 3) | (3, 3) | (3, 3) |
| Gaussian/uniform noise | $\sigma$ | $\frac{10}{255}$ | $\frac{20}{255}$ | $\frac{30}{255}$ | $\frac{40}{255}$ | $\frac{50}{255}$ | $\frac{60}{255}$ |
| Pixel dropout | $p$ | 0.1 | 0.2 | 0.3 | 0.4 | 0.5 | 0.6 |
| | | CIFAR-10 | | | | | |
| Gaussian blur | $\sigma$ | 1.0 | 2.0 | 3.0 | 4.0 | 5.0 | 6.0 |
| | $\kappa$ | (3, 3) | (3, 5) | (5, 5) | (5, 7) | (7, 7) | (9, 9) |
| Gaussian/uniform noise | $\sigma$ | $\frac{30}{255}$ | $\frac{60}{255}$ | $\frac{85}{255}$ | $\frac{100}{255}$ | $\frac{120}{255}$ | $\frac{140}{255}$ |
| Pixel dropout | $p$ | 0.1 | 0.2 | 0.3 | 0.4 | 0.5 | 0.6 |

Table 5: Overview of parameter values for the various image corruptions across datasets and intensity levels used for creating synthetic data for the evaluation of covariance shift detection in Section 5.2.

20 different models after training for 40 epochs, and models generally have different variance of weights and neural persistence values. However, we show that there is an approximately linear correspondence between the log-transformed variance and the neural persistence value. In Figure 11, we show results for models with relu activation function, and in Figure 12 we show results for models with tanh activation function. In each case, we show separate plots for models with different numbers of layers and trained on different datasets. Within each plot, we distinguish models with different hidden size by colours.

All plots agree with our observation that neural persistence corresponds approximately linearly to log-transformed variance. This can be seen from the resulting linear trajectories, each consisting of 20 points corresponding to the model variants with different initialisation and minibatch trajectory. Furthermore, our visual analysis is complemented by $R^2$ scores of linear regression fits, which we report in Table 6. All $R^2$ scores are close to 100%, which indicates that the log-transformed variance can explain a significant amount of variation of neural persistence values.

In summary, these results suggest that in the case of DNNs, neural persistence becomes a surrogate measure of the variance of weights.

### H.2 Variance and deep graph persistence as early stopping criteria

Rieck et al. (2019) suggest to use the growth of neural persistence values as an early stopping criterion. This is very useful in cases where no validation data is available. Therefore, we compare the variance and deep graph persistence as early stopping criterion to neural persistence.

Concretely, we evaluate numbers of warm-up epochs (referred to as "burn-in" epochs in (Rieck et al., 2019)) from 0 to 25 and use patience values between 1 and 10. We use Algorithm 2 in (Rieck et al., 2019) for calculating early stopping epochs from a maximum of 40 training epochs. Like (Rieck et al., 2019), we evaluate the model every quarter epoch.

Table 7 shows that using the variance as early stopping criterion, on average, stops around 2 epochs later than using neural persistence or the validation error as early stopping criterion. However, this actually leads to a slight expected increase of test set accuracy, both compared to neural persistence and to the validation

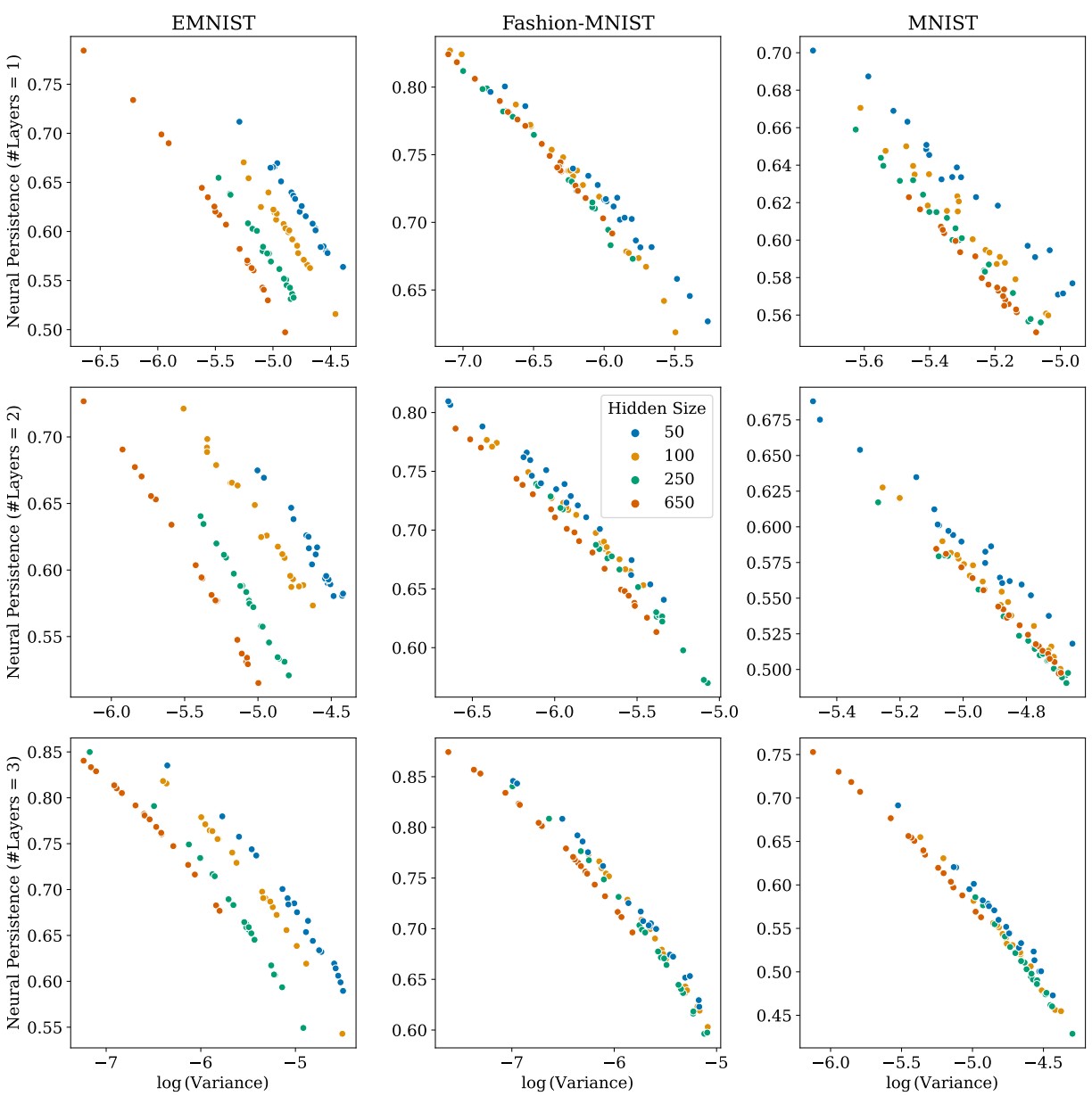

Figure 11: Roughly linear correspondence of log-variance of weights and neural persistence for models with ReLU activation function. Results are factorised by dataset (columns), number of layers (rows) and hidden size (colour).

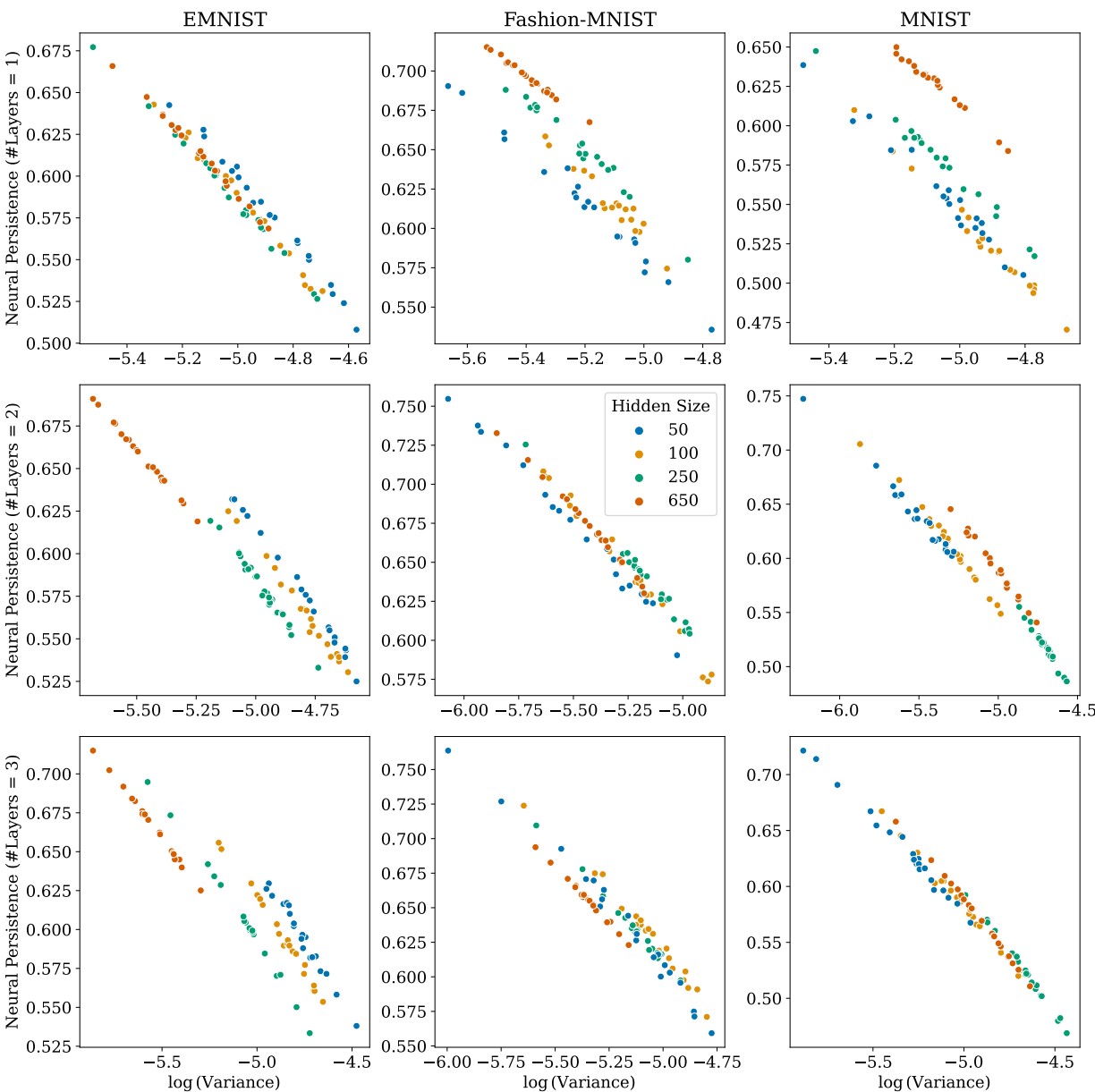

Figure 12: Roughly linear correspondence of log-variance of weights and neural persistence for models with tanh activation function. Results are factorised by dataset (columns), number of layers (rows) and hidden size (colour).

| | Dataset | EMNIST | | Fashion-MNIST | | MNIST | |
|---|---|---|---|---|---|---|---|
| | Activation | relu | tanh | relu | tanh | relu | tanh |
| #Layers | Hidden Size | | | | | | |
| 1 | 50 | 98.63 | 99.21 | 98.52 | 98.37 | 97.87 | 97.42 |
| | 100 | 97.56 | 99.03 | 97.98 | 93.27 | 96.95 | 99.10 |
| | 250 | 99.23 | 99.52 | 99.01 | 97.68 | 98.93 | 98.83 |
| | 650 | 98.68 | 99.68 | 99.39 | 99.39 | 98.82 | 98.93 |
| 2 | 50 | 96.44 | 99.39 | 97.72 | 98.87 | 98.24 | 98.47 |
| | 100 | 98.83 | 99.06 | 99.40 | 98.67 | 98.91 | 99.01 |
| | 250 | 99.57 | 98.88 | 99.34 | 99.02 | 99.48 | 98.37 |
| | 650 | 99.47 | 99.85 | 99.40 | 99.50 | 99.58 | 99.30 |
| 3 | 50 | 98.30 | 97.65 | 98.50 | 98.19 | 98.94 | 99.01 |
| | 100 | 98.16 | 99.15 | 99.52 | 98.04 | 99.09 | 99.26 |
| | 250 | 97.11 | 99.54 | 98.52 | 99.51 | 99.69 | 99.54 |
| | 650 | 98.67 | 99.73 | 98.64 | 99.61 | 99.36 | 99.72 |

Table 6: $R^2$ score of linear regression fits when treating the log-transformed variance of weights as independent variable and the neural persistence value as dependent variable. We show results for each model architecture, accordingly the linear regression is fitted on 20 datapoints that represent the 20 different runs with different initialisation and minibatch trajectory. All values are close to 1 (here scaled by factor 100 for better readability), which suggests excellent fit of the linear regression.

error. Also note that the difference in test set accuracy between neural persistence and variance is lower than the difference between neural persistence or variance and using the validation error as early stopping criterion. This shows that using the variance of weights can indeed be used as a less costly replacement of neural persistence as early stopping criterion, with a small expected increase in training time.

Similarly, Table 8 shows that using deep graph persistence as early stopping criterion does not yield notably different results compared to using neural persistence or the variance of weights as early stopping criterion. Averaged over all models and early stopping setting (i.e. number of warm-up epochs and patience), deep graph persistence stops earlier than neural persistence and variance of weights, but reaches slightly lower accuracy. However, given the generally small differences in accuracy, this does not constitute a qualitative difference.

Finally, in Figure 13, we show heatmaps in the style of Figure 4 in (Rieck et al., 2019). These particular results where obtained from 20 MLPs with 3 layers, 250 hidden units, and relu activation function trained on Fashion-MNIST. While we have made attempts to reproduce the results of Rieck et al. (2019) by contacting the authors, following their suggestions, and using their code to calculate neural persistence, we were unable to do so. We hypothesise that differences between our results and those presented in (Rieck et al., 2019) are due to differences in the trained models, for example different hyperparameter choices.

Our conclusion from these findings is that whether the generalisation capabilities of a model can be assessed from the parameters alone, i.e. without using any data, remains unclear.

### H.3 Variance and regularisers

We demonstrate that variance yields similar results compared to neural persistence for distinguishing deep feed-forward neural networks trained using different regularisers, namely dropout, batch normalisation, or no regulariser. To this end, we replicate the results from Section 4.1 in (Rieck et al., 2019). We train models using the same hyperparameters (3 layers, 650 hidden units, relu activation function) as Rieck et al. (2019) using no regulariser, batch normalisation, or dropout ($p = 0.5$) on the MNIST dataset. For each setting we train 30 models with different initialisations and minibatch trajectories.

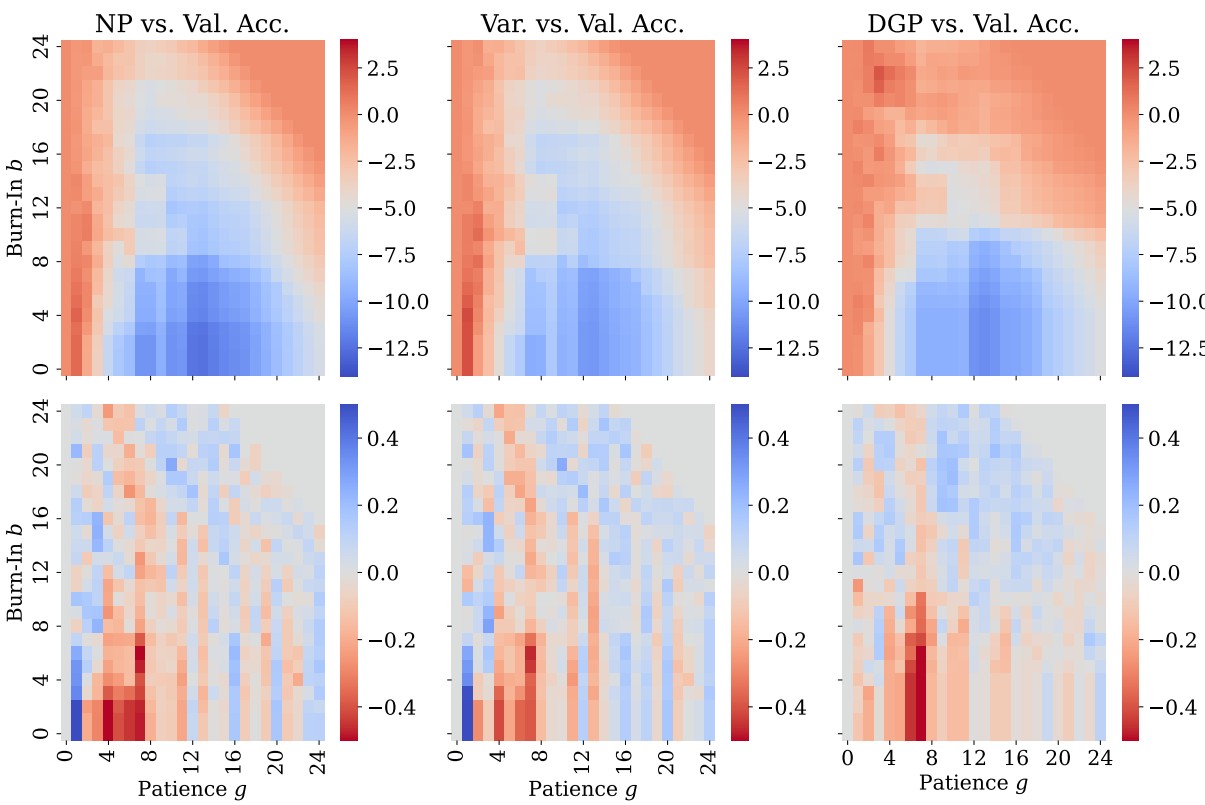

Figure 13: Differences between using accuracy on the validation set and neural persistence, variance, and deep graph persistence as early stopping criteria. The bottom row shows differences in test set accuracy at the point of early stopping, and the top row shows differences in the numbers of epochs. Numbers are read "value for first metric" minus "value for second metric".

|  | EMNIST | Fashion-MNIST | MNIST |
|---|---|---|---|
| $\Delta_{\text{epoch}}(\text{NP, val})$ | -0.91 | 0.68 | 1.76 |
| $\Delta_{\text{epoch}}(\text{NP, var})$ | -2.33 | -1.31 | -1.60 |
| $\Delta_{\text{epoch}}(\text{var, val})$ | 1.43 | 1.99 | 3.35 |
| $\Delta_{\text{accuracy}}(\text{NP, val})$ | 0.190 | 0.032 | 0.033 |
| $\Delta_{\text{accuracy}}(\text{NP, var})$ | -0.014 | -0.019 | -0.002 |
| $\Delta_{\text{accuracy}}(\text{var, val})$ | 0.204 | 0.050 | 0.034 |

Table 7: Differences in number of trained epochs when using validation error, neural persistence, or the variance of weights as early stopping criterion. $\Delta_{\text{epoch}}$ refers to the difference in number of trained epochs. $\Delta_{\text{accuracy}}$ refers to the difference in test set accuracy at the point of stopping. A positive difference means that the corresponding value for the first argument is higher. Note that accuracy is given in %, i.e. takes values in $[0; 100]$.

|                                  | EMNIST | Fashion-MNIST | MNIST  |
|----------------------------------|--------|---------------|--------|
| **Epoch difference**             |        |               |        |
| $\Delta_{\text{epoch}}(\text{DGP, val})$ | -1.945 | 0.711 | -0.490 |
| $\Delta_{\text{epoch}}(\text{DGP, var})$ | -3.376 | -1.28 | -3.348 |
| $\Delta_{\text{epoch}}(\text{DGP, NP})$  | -1.037 | 0.032 | -2.245 |
| **Test accuracy difference**     |        |               |        |
| $\Delta_{\text{accuracy}}(\text{DGP, val})$ | 0.189 | 0.034 | 0.027 |
| $\Delta_{\text{accuracy}}(\text{DGP, var})$ | -0.015 | -0.017 | -0.007 |
| $\Delta_{\text{accuracy}}(\text{DGP, NP})$  | -0.002 | 0.002 | -0.006 |

Table 8: Differences in number of training epochs when using validation error ("val"), neural persistence ("NP"), or variance ("var") as early stopping criterion compared to using deep graph persistence ("DGP") as early stopping criterion. $\Delta_{\text{epochs}}$ refers to the difference in number of training epochs. $\Delta_{\text{accuracy}}$ refers to the difference in test set accuracy at the point of stopping. A positive difference means that the corresponding value for the first argument is higher. Note that accuracy is given in %, i.e. takes values in $[0; 100]$.

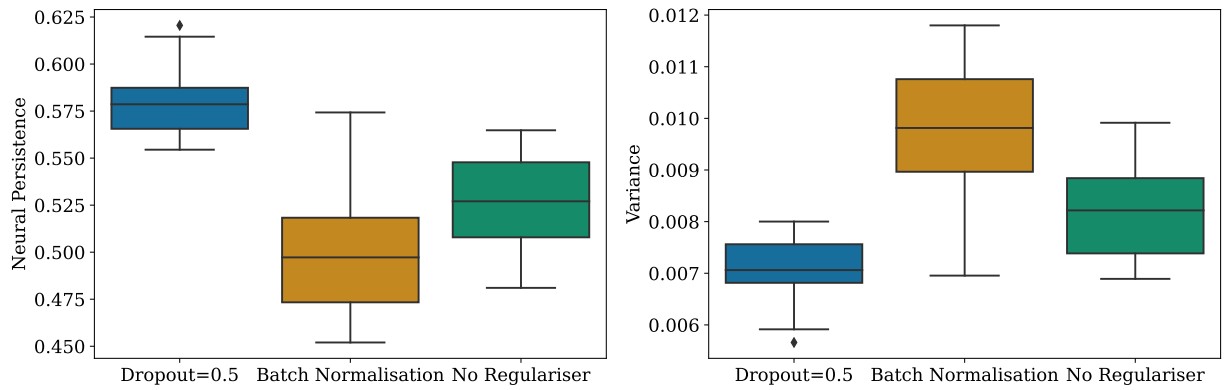

Figure 14: Left: Replication of Figure 3 in (Rieck et al., 2019). Neural persistence can distinguish models trained on MNIST using different regularisers. Right: Using variance of weights instead of neural persistence yields very similar results. Note, that results for variance of weights are vertically flipped compared to neural persistence, because higher neural persistence corresponds to lower variance of weights.

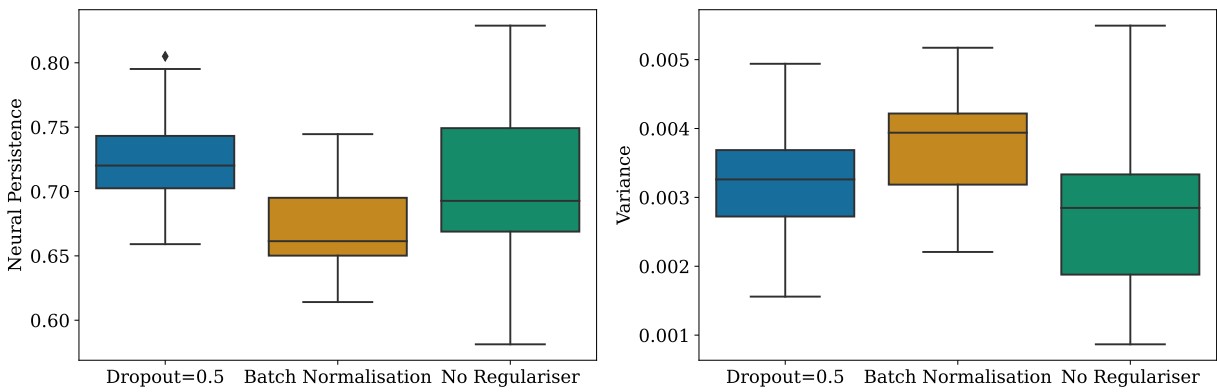

Figure 15: Using neural persistence or variance of weights fails to distinguish models trained on Fashion-MNIST using different regularisers. However, distributions for variance of weights and neural persistence are again similar up to vertical flipping.

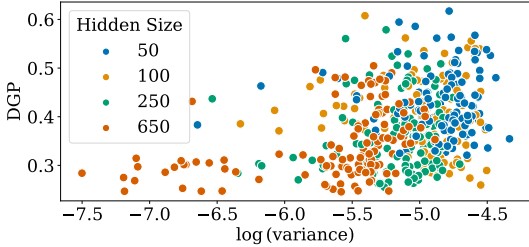
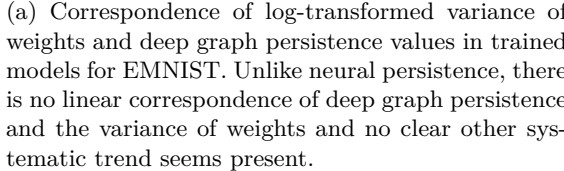
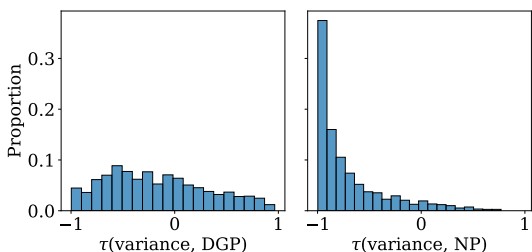

(a) Correspondence of log-transformed variance of weights and deep graph persistence values in trained models for EMNIST. Unlike neural persistence, there is no linear correspondence of deep graph persistence and the variance of weights and no clear other systematic trend seems present.

(b) Distribution of Kendall's $\tau$ between variance of weights and deep graph persistence values over training (right) and neural persistence values over training (left). While neural persistence and the variance of weights are highly correlated in most cases, there is no clear trend regarding strong correlation of deep graph persistence and variance of weights.

Figure 16: DGP does not correspond linearly to the variance of weights.

Then, we compute the neural persistence and variance of normalised weights for each model. Figure 14 shows that the distribution of variances differs between models trained with different hyperparameters. Results for neural persistence are the similar. On the one hand, this reproduces the results in (Rieck et al., 2019), but confirms that variance behaves similar to neural persistence in this regard as well.

Finally, however, we want to mention that we believe that these results are actually an artifact of this particular dataset, namely MNIST. Figure 15 shows the corresponding results for models trained on the Fashion-MNIST dataset. Here, we do not see any clear distinction between distributions of variances or neural persistence values for different datasets.

### H.4 Dependence of deep graph persistence on variance.

We verify that deep graph persistence does not correlate strongly with the variance of weights of neural networks unlike neural persistence. Similar to the analysis in Section 4.2, we analyse the relation between deep graph persistence and the log-transformed global variance of all weights in a trained network.

In Figure 16b, we show the distribution of correlation coefficients (Kendall $\tau$) of deep graph persistence and variance and of neural persistence and variance during training. While in the vast majority of cases neural persistence and variance evolve similarly during training, the distribution of correlation coefficients is generally flat. This suggests that there is no systematic co-evolution of deep graph persistence and the variance of weights during training.

In Figure 16a, we show that there are no visible trends regarding correspondence of deep graph persistence and variance in trained neural networks. Here, we use the same set of models that we used in Section 4. Also, we evaluate models after training for 40 epochs. Therefore, we conclude that unlike neural persistence, deep graph persistence cannot be replaced by only considering the variance of weights.

### H.5 Additional results for deep graph persistence on covariate shift detection

In Figure 17, we show the detection ratios of all methods in a visual manner for the three different datasets. Here, the corruption is Gaussian noise. Figure 17 shows that deep graph persistence outperforms the baselines with a particularly large margin on CIFAR-10, which is the most realistic of the three datasets. On Fashion-MNIST, the distance to the softmax baseline is small, however. In Figure 18, we show how the different feature extraction methods are affected by only having a certain ratio $\delta$ of test samples (from the CIFAR-10 data) being corrupted (by Gaussian noise). In Table 1 and Figure 17, detection ratios for the different corruption ratios are aggregated. Figure 18 shows that batches containing fewer corrupted images are harder

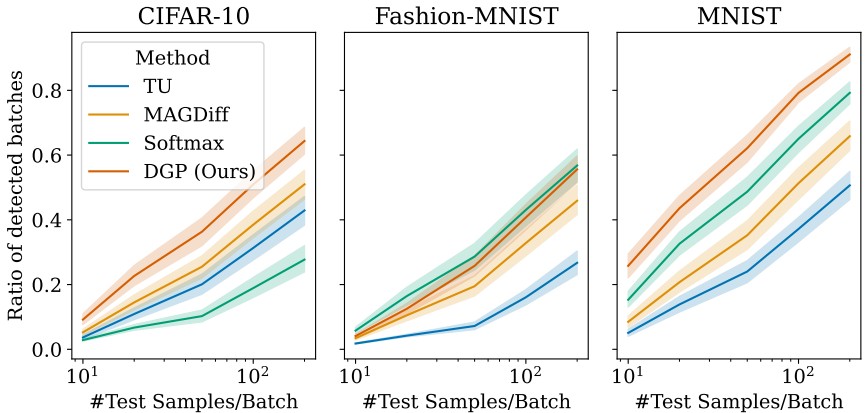

Figure 17: Corruption detection ratios for different sample sizes and feature extraction methods shown for different datasets. Here, the corruption is Gaussian noise.

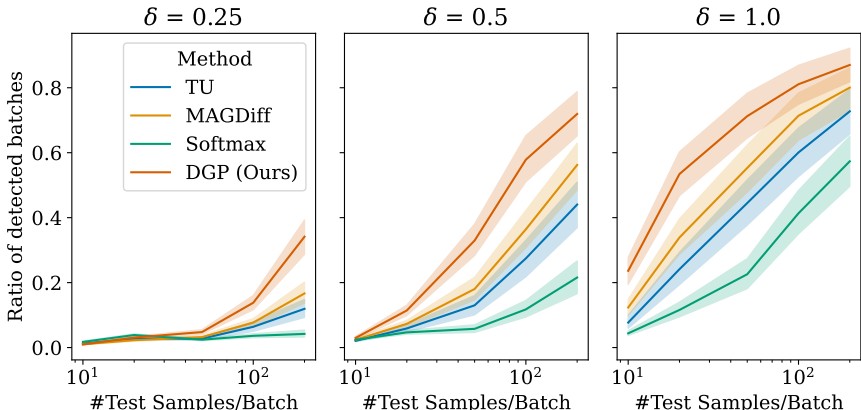

Figure 18: Corruption detection ratios on CIFAR-10 for different sample sizes and feature extraction methods for different ratios of corrupted test batches (e.g. $\delta = 0.25$ means that 25% of images in the test batch are corrupted, while all other images are unaltered.) The corruption used is Gaussian noise.

to detect, as expected. But for $\delta = 0.25$, from a certain sample size on, the detection ratio of deep graph persistence increases rapidly, while that of baselines does not increase as fast. For higher values of $\delta$, all methods can detect corrupted batches given a sufficiently large sample size $n$.

## H.6 Full results of deep graph persistence ablation on MNIST

In Table 9 we report full results of our ablation study for deep graph persistence on MNIST. In the main part, we only reported sample sizes $n \in \{20, 50, 100\}$. Here, we additionally report results for sample sizes 10 and 200. These results do not provide additional information that are not visible for the sample sizes reported in the main part.

## H.7 Deep graph persistence yields calibrated detection scores

In Table 10, we show that both TU and deep graph persistence yield calibrated detection scores for detecting corrupted images. Specifically, we show that false detection rates, i.e. returning that a batch of images contains corrupted samples when actually all are not corrupted, is small. This is important to verify that high true detection rates are not simply caused by too high sensitivity to spurious differences in feature distributions. Following Hensel et al. (2023), we take a false detection rate of 5% as an upper threshold for

| $n$ | | TU | TU (+ norm) | DGP (− norm) | DGP |
|---|---|---|---|---|---|
| 10 | GB | 1.63 | **1.93** | 1.66 | 1.91 |
| | GN | 5.07 | 23.41 | 18.82 | **25.79** |
| | PD | **17.03** | 15.43 | 14.05 | 14.90 |
| | UN | 2.27 | 14.46 | 9.81 | **16.96** |
| 20 | GB | 3.85 | **4.94** | 3.66 | 4.92 |
| | GN | 13.85 | 40.55 | 35.15 | **43.57** |
| | PD | **31.02** | 28.51 | 27.72 | 28.44 |
| | UN | 6.54 | 28.34 | 22.46 | **32.56** |
| 50 | GB | 6.34 | 9.31 | 5.15 | **9.71** |
| | GN | 24.02 | 57.44 | 51.78 | **62.13** |
| | PD | **46.74** | 43.40 | 41.15 | 43.20 |
| | UN | 12.28 | 43.23 | 35.70 | **48.92** |
| 100 | GB | 12.35 | 18.48 | 11.26 | **20.16** |
| | GN | 37.17 | 75.06 | 67.71 | **79.21** |
| | PD | **62.33** | 58.26 | 55.95 | 58.14 |
| | UN | 21.31 | 57.80 | 49.74 | **64.71** |
| 200 | GB | 19.38 | 28.45 | 19.80 | **30.19** |
| | GN | 50.64 | 86.74 | 81.76 | **91.08** |
| | PD | **75.64** | 71.39 | 68.24 | 71.58 |
| | UN | 31.92 | 70.79 | 63.27 | **78.71** |

Table 9: Full ablation results for modifications of neural persistence made to obtain deep graph persistence. "± norm" denotes whether layer-wise standardisation is applied. Scores are detection ratios on MNIST reported as percentages in range $[0, 100]$.

| $n$ | Method | CIFAR-10 | Fashion-MNIST | MNIST |
|---|---|---|---|---|
| 10 | TU | 1.04 | 1.23 | 1.09 |
| | DGP (Ours) | 0.89 | 1.15 | 1.09 |
| 20 | TU | 1.89 | 2.25 | 1.97 |
| | DGP (Ours) | 1.55 | 2.09 | 1.98 |
| 50 | TU | 1.17 | 1.28 | 1.31 |
| | DGP (Ours) | 0.98 | 1.25 | 1.19 |
| 100 | TU | 1.12 | 1.29 | 1.13 |
| | DGP (Ours) | 1.00 | 1.30 | 1.05 |
| 200 | TU | 0.56 | 0.62 | 0.58 |
| | DGP (Ours) | 0.47 | 0.62 | 0.56 |

Table 10: False detection rates for TU and deep graph persistence on different datasets. Numbers are reported in percent, i.e. in $[0; 100]$. All scores are $< 5\%$ which means that methods only rarely mistakenly return that a batch of images contains contains corrupted samples when in reality all images are not corrupted.

well-calibrated detection scores. Numbers in Table 10 clearly indicate that false positive rates for both TU and deep graph persistence are well below 5%. Therefore, we can be confident that deep graph persistence actually captures differences in distribution between corrupted and non-corrupted images.

