# OpenReview forum: "Addressing caveats of neural persistence with deep graph persistence"
_TMLR — Accepted by TMLR_

### Review · Reviewer_KwHa · 2023-08-10

**Summary Of Contributions:**

The contributions of this paper are two-fold:

First, [Neural Persistence](https://openreview.net/forum?id=ByxkijC5FQ),
a topology-driven measure for assessing the topological complexity of
fully-connected neural networks, is analysed and compared to the
variance of *neural network weights*. The implication of this is that,
as the neural network becomes deeper, the changes in neural persistence
are rather driven by the *variance* of the weights as opposed to any
structure in the weights themselves. This is shown to imply a linear
relationship between the variance of weights and neural persistence.
Interestingly, these findings do *not* translate to deeper neural
networks.

Second, the paper introduces a way to address the shortcomings of neural
persistence. This is achieved by generalising the approach of neural
persistence to deep neural networks, taking into account _all_ weights
as opposed to the weights of only a single layer. This extension is
shown to result in a method that is more aligned with the complexity of
the neural network representation, and is overall less susceptible to
mere changes in the variance.


**Audience:**

Yes

**Broader Impact Concerns:**

There are no broader impact concerns to be raised by this work.


**Claims And Evidence:**

Yes

**Requested Changes:**

- The discussion on 'measure' in the sense of 'measure theory' (footnote 1) strikes me as redundant. Consider removing it.

- When discussing *topological uncertainty* for the first time, please
  add a citation.

- Please comment briefly on why the work of 'Liu et al. (2020)' is not
  suitable as a replacement of the proposed method. It seems that this
  work would also enable the experiments proposed in the paper at hand.

- Some of the citations to 'Rieck et al. (2019)' strike me as redundant,
  in particular when they appear in close sucession. For instance, in
  the Section 3, there are numerous sentences that end with such
  a citation. Consider rephrasing some of these sentences.

- In Definition 3.2, please add 'graph' (instead of just saying
  'Complete bipartite').

- Please comment briefly on whether higher-order topological features
  considered for each layer would address some of the problems
  mentioned in this work. This would substantially increase our
  knowledge on the utility of per-layer scores of the complexity of
  neural networks.

- Please expand on the idea of spatial structure in a matrix.
  I understand the gist of it, but it seems like a non-standard
  definition that, moreover, is susceptible to permutations (in the
  sense that the indexing of features, which is somewhat arbitrary)
  seems to matter.

- Please comment briefly on whether a single-layer network can be
  considered *linear*; are you not making use of non-linear activation
  functions, which influence weights?

- When presenting the results in Section 4.3, consider using the
  visualisations employed by Rieck et al., to make the results directly
  comparable.

- Personally, I would prefer showing Section 5 much *earlier* in the
  paper and present one **joint** section for experiments that serve to
  show their improvements.

- Consider rephrasing the discussion on computational complexity for the
  new measure. Since this entails finding a set of weights across *all*
  layers, a more detailed complexity would be appreciated.

## Minor issues

- 'analogue fashion' --> 'analogous fashion'

- Please check the references for consistency, in particular when it
  comes to capitalisation of proper names like MNIST.

## Summary

Overall, this paper has the potential to substantially impact the
literature in the nascent field of topological machine learning.
I believe that additional rephrasing to **improve the clarity** are
sufficient to turn this into a great publication.


**Strengths And Weaknesses:**

The primary strength of the paper is that it is operating on sound
mathematical principles and providing a well-justified critique of an
existing method. The analysis proceeds carefully and in an overall sound
fashion; care is taken to motivate the individual steps and intermediate results.

At the same time, the current write-up is suffering from issues of
**clarity**, at least in the expository sense: while the materials
discussed are without a doubt highly relevant, there is the question of
why so much time is spent discussing the shortcomings of neural
persistence, when it would be easier to just 'pitch' the modified
version and use that in lieu of the original version. Moreover, the
experimental setup, as far as this reviewer can tell, does not follow
the original paper on [neural
persistence](https://openreview.net/forum?id=ByxkijC5FQ). A connection
to the scenarios outlined by Rieck et al. would be most desirable,
though, since it would make it easier for non-expert readers to
understand the novel claims.

---

> ### Author Response · Authors · 2023-09-21
> **Thank you for the review.**
>
> We would like to thank the reviewer for the useful and detailed feedback, including their appreciation of our ‘well-justified critique of an existing method’ which includes the presentation of a ‘careful’ and ‘sound’ analysis. Below, we reply to all the requested changes by the reviewer and detail how we incorporated those in our updated manuscript.
>
> > ### **The discussion on 'measure' in the sense of 'measure theory' (footnote 1) strikes me as redundant. Consider removing it**
>
> The footnote was removed in the updated manuscript.
>
> > ### When discussing topological uncertainty for the first time, please add a citation.
>
> We added a citation on page 2, i.e. in the Introduction.
>
> > ### **Relation between DGP and ‘Liu et al. (2020)’**
>
> Indeed, the definition of $\langle \epsilon=\phi\rangle$-connected (Definition 3) in (Liu et al., 2020) when applied only to the input and output layers, effectively leads to the same filtration as our proposal for DGP. Therefore, transferring this to the setup in Section 5 in our paper corresponds to the case “DGP (-norm)” which is evaluated in Section 5.3. However, this variation of DGO yields significantly weaker results than our proposed formulation for most evaluated image corruptions, as shown in Table 2. Therefore, we conclude that the layer-wise re-normalisation as in Equation (9) is an important ingredient to boost performance.
>
> In our revised manuscript, we added a discussion about the connection between DGP and (Liu et al., 2020) in Section 5.
>
> > ### **Redundant citations of ‘Rieck et al. (2019)**
>
> In the revised manuscript, we have changed / removed the following mentions of ‘Rieck et al. (2019)’:
>   *  The second mention in the introduction was changed to ‘Rieck et al.’ (\citeauthor).
>   *  The mention in the second paragraph in Section 3 was removed.  We left the remaining two mentions of ‘(Rieck et al. 2019)’ in Section 3 unchanged, because they serve the important purpose of referring to theorems from their paper, and not those in our paper. The mention of ‘Rieck et al.’ in paragraph “Normalised neural persistence.” in Section 3 was removed.
>   *  All mentions in Section 4.1 except for two mentions of ‘Rieck et al.’ in places where no unambiguous replacement is possible were removed.
>   * The mention in Section 4.3 was changed to ‘Rieck et al.’ (\citeauthor).
>   * The two mentions in Section 5.1 were changed to ‘Rieck et al.’ (\citeauthor).
>
> We left the references in the related work section unchanged.
>
> > ### **In Definition 3.2, please add 'graph' (instead of just saying 'Complete bipartite')**
>
> This was added in the revised manuscript.
>
> > ### **Discuss the utility of higher order topological features for the evaluated tasks**
>
> Since we represent neural networks as graphs, we obtain a data structure that itself is a 1-dimensional simplicial complex. Therefore, we can only obtain topological features of dimension $> 1$ by considering clique complexes (or path homology (Chowdhury and Mémoli, 2018), for which however to our knowledge no efficient computational methods exist). However, as already noted by Rieck et al. and other authors, cliques in layers of neural networks do not have a conceptually clear meaning. For example, consider cycles (1-dimensional features): Intuitively, computations in MLPs are unidirectional in the sense that information is passed from the input layer to the output layer. Then, the relation between any cyclic structure established based on the weights and the actual function represented by the network is unclear. Similar arguments can be made for higher-order topological features.
>
> [1] Chowdhury, Samir, and Facundo Mémoli. "Persistent path homology of directed networks." In: Proceedings of the Twenty-Ninth Annual ACM-SIAM Symposium on Discrete Algorithms. 2018.
>
> > ### **Clarification of the term ‘spatial structure (in a matrix)’**
>
> We have updated Section 3 (pages 4 and 5) to enhance clarity regarding the spatial structure of a matrix (Section 3).
>
> In particular, we consider if large entries concentrate in particular rows or columns (or only certain combinations of rows and columns). This intuition does not depend on any particular ordering of rows or columns because permuting them does not change whether large entries concentrate in some rows (or columns). However, in practice it is difficult to detect spatial concentration. In Appendix E, we discuss how to measure spatial concentration and how to artificially construct matrices with pre-defined spatial concentration.
>
> For our empirical experiments in the main paper, we try to avoid explicitly dealing with spatial concentration and only measure the effects of shuffling matrix entries which goes beyond permuting (entire) rows and columns. Shuffling destroys any spatial structure, and so we can analyse if neural persistence depends only on which entries appear in a matrix, or if it also, as we argue, depends on where entries appear in the matrix.

---

> > ### Author Response · Authors · 2023-09-21
> > **Thank you for the review (continued).**
> >
> > > ### **Clarification of the term ‘(linear) layer’**
> >
> > We are sorry for the unclear terminology regarding the definition of a ‘layer’. With ‘layer’, we refer to the linear transforms, i.e. weight matrices, in linear and deep models. In this case, a linear classifier contains one layer, i.e. one weight matrix, which directly maps the input features to the class logits. A deep network with a single nonlinearity contains two layers, one layer for mapping the input units to the hidden representations, and one mapping the hidden representations to the class logits. We have unified our terminology throughout the revised manuscript and we hope that this has raised clarity to an acceptable level.
> >
> > > ### **Present figures like those in (Rieck et al., 2019) in Section 4.3**
> >
> > The main part of our analysis, i.e. how neural persistence in practice relates to variance, was not considered in (Rieck et al., 2019). Therefore we cannot provide comparable visualisations for those.
> >
> > We replicated Figure 3 from (Rieck et al., 2019) and show that variance and NP yield similar capabilities for distinguishing regularizers (Figures 14 and 15 in the Appendix).
> >
> > Regarding Figure 4 in (Rieck et al., 2019), the authors could unfortunately not provide their code for recreating the figure. We followed their suggestions, but were not able to obtain a similar figure. We include our reproductions in Appendix H.2 (Figure 13), which however differ from those reported in (Rieck et al., 2019).
> > In particular, we used the code accompanying (Rieck et al., 2019) to calculate neural persistence, therefore differences between our results and those presented in (Rieck et al., 2019) are likely due to differences in the trained models, e.g. hyperparameters, and are not caused by different implementations of the NP calculation.
> >
> > > ### **Re-structuring the paper to increase emphasis on Section 5**
> >
> > We agree that prominently showing strong results in an empirical setting is desirable and would make the paper more interesting to readers who want to apply topological methods. We consider it a great merit of Rieck et al. (2019) to enable practical applications of topological methods in the machine learning context.
> >
> > However, our paper mainly focuses on the theoretical analysis of neural persistence. Our intention of presenting the results in Section 5 is to show that some aspects of the correspondence between neural persistence and variance can be addressed. Our updated method (DNP) is useful for practical tasks, albeit not in the setting originally discussed in Rieck et al. (2019), and we therefore see DGP as an additional contribution of our work, rather than the main contribution.
> >
> > > ### **Discuss the computational complexity of calculating DGP**
> >
> > We have updated Section 5.1 (page 9) to include a discussion of the computational complexity of calculating DGP. Furthermore, we added pseudocode of the algorithm for calculating the summary matrix $S$ (Appendix F in the revised manuscript).
> >
> > Specifically, calculating DNP is composed of representing the network as a single bipartite graph and then calculating the MST. Calculating the MST with Kruskal’s algorithm is $\mathcal{O}(V^2 \cdot \log V)$, where $V$ is the sum of the number of input features and the number of output features. The computational complexity of representing the graph as a single matrix is $\mathcal{O}(L \cdot o \cdot d^2)$, where $L$ is the number of layers, $o$ is the number of output units (usually $\ll d$), and $d$ is the hidden size.

---

> > > ### Comment · Reviewer_KwHa · 2023-09-29
> > >
> > > Dear authors,
> > >
> > > Thank you for an outstanding update of the paper. I believe that all my concerns have been addressed, and I agree with keeping the structure in place as-is. I believe the paper is indeed almost ready for publication; I have but one final remark or change request concerning the reproducibility aspects: in your rebuttal, you hypothesise that the issues pertain to hyperparameter choices. Please update the phrasing you use in the current revision to include this aspect; the current phrasing is _ambiguous_ in the sense that readers do not have access to the correspondence with the authors and do not know about the precise nature of the efforts to reproduce their results. **I would thus suggest a more neutral phrasing, such as**:
> > >
> > > > While we have made attempts to reproduce the results of [...], we were unable to do so. We hypothesise that [...]

---

> > > > ### Author Response · Authors · 2023-09-30
> > > >
> > > > Thank you for the positive assessment of our revision. We are happy that we could address all your concerns. Also, thank you for your valuable suggestion. We acknowledge that the current phrasing could be interpreted in a negative fashion we did not intend. We have updated the phrasing to be more neutral, following your proposal. Thank you for noticing this issue.

---

> > > > > ### Comment · Reviewer_KwHa · 2023-10-01
> > > > >
> > > > > Thank you very much for being responsive. In my opinion, the paper is now indeed ready for publication.

---

### Review · Reviewer_zwz9 · 2023-08-25

**Summary Of Contributions:**

The authors study the neural persistence framework, which previously proposed to exploit TDA principles to assess neural network complexity. In spite of neural persistence's influence, the authors unearth the key factors steering the persistence measure, finding them to correspond to the variance of the weights in the shallow layers of the model. Hence, it is concluded that the neural persistence as presented is limited, and a more global variant of it, dubbed "graph persistence" is proposed. Theory and experiments supplement the value of the graph persistence measure.

**Audience:**

Yes

**Broader Impact Concerns:**

No concerns.

**Claims And Evidence:**

Yes

**Requested Changes:**

See the second paragraph in 'Strengths and Weaknesses'.

**Strengths And Weaknesses:**

While I am not an expert in the neural persistence literature, I find the work to be extremely valuable and timely. Identifying the limitations of a prior top-tier paper, both empirically and theoretically, is a relatively rare sight in deep learning research, and I applaud the authors for making such a result possible! The proposed modifications are sensible and lead to improvements on the regimes studied. I would recommend acceptance even in the current form of the work.

My main concern is centered around the downstream impact of the novel graph persistence measure. As mentioned in the paper, one of the 'selling points' of neural persistence is its valuable ability to early-stop without a validation set; however, the results in this paper to that end turn out to be negative. While there is appropriate referencing around this result, I would have appreciated a more deeper (empirical or theoretical) analysis on why the deep graph persistence measure was unable to serve as a better proxy for early stopping. In my opinion, such a result, or any other additional result indicating how a downstream neural network practitioner could make effective use of this function, would be highly appreciated.

---

> ### Author Response · Authors · 2023-09-21
> **Thank you for the review.**
>
> We would like to thank the reviewer for their very positive review, characterising our work as ‘extremely valuable and timely’. We appreciate the reviewer’s positive comment about our analysis of previous work being ‘a relatively rare sight in deep learning literature’ along with their suggestion of ‘acceptance even in the current form of the work’.
> Below, we reply to the concerns raised by the reviewer and include additional experiments for analysing the use of NP and DGP as early stopping criteria.
>
> > ### **Deeper analysis of why DGP does not present a better early stopping criterion than NP**
>
> We conducted additional experiments to evaluate the behaviour of DGP (and NP) in a setting with extreme overfitting, i.e. where the validation error not only stagnates, but also increases. Most importantly, these experiments show that DGP does not adequately reflect generalisation error.
>
> More specifically, in the considered scenario the validation error begins to increase again relatively early in training, but eventually converges to a somewhat higher value than the minimum reached early in training. However, neither neural persistence nor deep graph persistence adequately reflect these changes: Neural persistence rises monotonically for a long time and then stagnates. This means that the evolution of neural persistence values neither indicates the start of overfitting, nor the final convergence of the model. Deep graph persistence shows a less smooth evolution, but structurally it evolves similar to neural persistence. Deep graph persistence also reaches its global maximum around when neural persistence starts to stagnate and then starts to slowly decrease.
>
> We decided to not include these experiments as an Appendix in the revised manuscript, because we doubt they are interesting to all readers. However, if the reviewer deems it useful, we would be happy to also include these new experiments.
>
> As suggested by our experiments, we believe that the failure of DGP to present a useful early stopping criterion is largely due to it only measuring properties of the model’s parameters, but not properties of the function represented by the network.
>
> This being said, NP shares the same problems. In general, neural persistence, variance, and also deep graph persistence when applied to the (static) weights are only measures of the fixed parameters of the network. However, generalisation is a characteristic of the function realised by the neural network that is parametrized by the parameters at hand. It is important to make the distinction between the function represented by the neural network and the network’s parameters.
>
> Now, we can refer to other works that have investigated the relation between parameters and the function. Most importantly, Benjamin et al. (2018) have conducted such an analysis. Their results suggest that the relation between parameters and function is not really transparent and certainly not linear.
>
> Therefore, our first answer to the question is that measures that only look at parameters are unlikely to capture all nuances of the network in function space. This is also supported by large-scale experiments in generalisation research, such as (Jiang et al., 2019), where parameter norm based generalisation bounds are found to be comparatively ineffective, whereas sharpness-based measures that look at the relation between changes in parameter space and the resulting changes of the loss value are found to be more successful.
>
> Our interpretation of these findings is that both measures only reflect global structural properties, such as variance or ratio of maximum weight over mean weight, but not more nuanced functional changes.
>
> We therefore conclude from our observations and the insights found in related literature, that it is important to also consider properties of the function represented by the neural network, and not only its parameters, as neural persistence originally proposed.
>
> While our proposed application of DGP on the activation graph in fact could achieve this, we think that this approach is too inefficient to be of practical relevance: Calculating activation graphs and DGP is computationally very costly for a larger number of samples and would significantly slow down training. Also, calculating the activation graph of large-scale models could be problematic in terms of memory consumption.
>
> In summary, we agree that a more detailed analysis of overfitting would be extremely interesting. Unfortunately, discussing the field of generalisation research is out of scope of what we originally intended with our work. We definitely hope we can expand on this from a topological data analysis perspective in future work.
>
> [1] Benjamin, Ari, David Rolnick, and Konrad Kording. "Measuring and regularizing networks in function space." In: ICLR. 2018.
>
> [2] Jiang, Yiding, et al. "Fantastic Generalization Measures and Where to Find Them." In: ICLR. 2019.

---

> > ### Author Response · Authors · 2023-09-21
> > **Thank you for the review (continued).**
> >
> > > ### **Request to present results showing the usefulness of DGP for downstream applications**
> >
> > We agree that finding more practical applications of neural persistence or deep graph persistence would greatly benefit the machine learning and topological data analysis fields, and therefore we are grateful for the encouragement to develop such applications in future work.
> >
> > In this paper, we consider one such application, namely the detection of corrupted images, that has already been discussed in previous work. This is a very useful application, because it enables the detection of domain shifts, for instance in scenarios where data arrives as a continuous stream. In case of changes in the underlying data-generating process, the newly arriving data points would exhibit different characteristics, and models trained on in-domain data may not work well anymore.
> >
> > Most of our experiments that evaluate whether neural persistence or deep graph persistence can be used to assess generalisation capabilities of models turned out to be negative, as already discussed above. This was already noted by Rieck et al., who in the appendix of their paper include their finding that neural persistence (after training) does not correlate with test set performance. We can confirm this finding. This suggests that applying topological data analysis to data, or representations of data, instead of models, is currently the most viable way to make use of these methods in practice.

---

> > > ### Comment · Reviewer_zwz9 · 2023-09-25
> > > **Thank you!**
> > >
> > > Thank you for carefully responding to my review. It would have been nice to show more downstream uses for the new measure, but I am concurring to leave that to the future work. I reaffirm my support for acceptance.

---

### Review · Reviewer_2V1F · 2023-09-17

**Summary Of Contributions:**

In this article, the authors demonstrate that the neural persistence (NP), which is a common tool for characterizing neural networks, is actually bounded by the network weight variance and spatial organization. Then they demonstrate that while NP varies a lot between these bounds for linear classifiers, this is not true for deep neural networks, where it is mostly correlated to the weight variance. Thus they propose to extend the NP definition to the whole network instead of averaging across layers, and show empirically that this new NP definition allows for better separation between models on shift detection experiments.

**Audience:**

Yes

**Claims And Evidence:**

Yes

**Requested Changes:**

+++ Section 4.2: why not running statistical tests and providing p-values (instead of showing box plots) for studying the effect of randomly shuffling weights, since there is a null model? This would be more convincing than displaying NP \Delta values that are not as compelling.

+++ Section 5.1: what is the computational cost for computing Equation (10)? It would be great to provide pseudo code and complexity analysis for DGP, since one of the advantages of the other methods (which average layers) is that they are at least quite fast to compute, so it would nice to show that the additional computational cost for DGP is not too large.

+++ Section 5.2: what is the rationale for representing persistence diagrams with vectors of sorted weights? Is it a stable representation (i.e., is the Euclidean distance between these representations upper bounded by the bottleneck distance between the diagrams themselves)?

+++ Appendix A: I would like to see more explanations for Remark A.7 as it is a bit vague for now. The way it is currently stated makes the derivation of Equation (20) quite unclear.

Typos:

p4: "we choose the value that maximises NP, i.e., 0": shouldn't it be 1?
p5: "U is tighter when $B_{\not\sim A}$" is smaller": shouldn't it be "|B\B_{\not\sim A}|"?
p5: "we have equality of NP_p(W) = U": shouldn't it be "(U^p-1)^{1/p}"?

**Strengths And Weaknesses:**

Overall I feel positive about this work. I think it presents interesting and convincing ideas, and is quite well-written and easy to read. To my opinion it is a useful contribution to the TDA community. I still have a few comments that I would like the authors to answer / discuss though.

---

> ### Author Response · Authors · 2023-09-21
> **Thank you for the review.**
>
> We would like to thank the reviewer for the positive and constructive feedback, describing our work as a ‘useful contribution to the TDA community’, and acknowledging that our work presents ‘interesting and convincing ideas’ and is ‘well written and easy to read’. In the following, we reply to all the requested changes by the reviewer and detail how we incorporated the feedback in our updated manuscript.
>
> > ### **Present p-values for experiments testing the presence of spatial concentration in weight matrices**
>
> As suggested, we performed a two-sample Kolmogorov-Smirnov test to investigate the null hypothesis if NP values of layers for different runs with the same hyperparameters are identically distributed as NP values resulting from permuting the entries in the respective weight matrices. In the case of linear classifiers, the KS-tests reject the null hypothesis (that NP values with and without permutation are equally distributed) with p-values $\ll 10^{-6}$. In the case of deep classifiers, the KS-tests reject the null hypothesis for the first layer (that is connected to the input features) in 80%-90% of cases and fail to reject the null hypothesis for all deeper layers, which supports our results.
> We have added those results in Section 4, in addition to the box plots (Fig. 1) which provide further visualisations for the presence of spatial concentration of large weights in linear classifiers.
>
> > ### **Discuss the computational complexity of calculating DGP**
>
> We have included a discussion of the computation complexity of DGP on page 9 and provide pseudocode for computing the summary matrix in Appendix F.
>
> The computational complexity of DGP is composed of the computational complexity for calculating the summary matrix $S$, which is $\mathcal{O}(L \cdot o \cdot d^2)$ where $L$ is the number of layers, $o$ is the number of output units, and $d$ is an upper bound for the hidden size and the number of input features, and the computational complexity of calculating NP for the summary matrix, which is $\mathcal{O}(V^2 \cdot \log V)$, where $V$ is the number of input units plus the number of output units.
>
> In many cases $L$ and $o$ will be very small, so that formally the computational complexity of calculating the layer-average NP value, which is $\mathcal{O}(L \cdot d^2 \cdot \log d)$, are similar. However, DGP is in practice faster, because the involved calculations are easily parallelizable and therefore more efficient on modern hardware.
>
> > ### **Rationale for representing persistence diagrams with vectors of sorted weights**
>
> The rationale for representing persistence diagrams with vectors of sorted weights is outlined in (Lacombe et al.. 2021), Section 2.2 (Computing and averaging persistence diagrams). We have added a reference on page 10.
>
> The main reason is that we want to compute the Wasserstein distance between the two persistence diagrams, which in our case reduces to comparing two equally sized samples of 1-dimensional distributions. There, the optimal matching is obtained by considering samples when sorted in increasing order.
>
> > ### **Clarification of Remark A.7**
>
> Thanks for pointing this out. We have added the inductive argument referred to in Remark A.7, and hope that this improved clarity.
>
> > ### **p4: "we choose the value that maximises NP, i.e., 0": shouldn't it be 1?**
>
> The term to compute NP has the form $\Sigma_w (1 - w)$, therefore the value for $w$ that maximises NP should be 0, not 1, as the sum would be 0 if all $w$ are 1.
>
> > ### **"U is tighter when " is smaller": shouldn't it be "|B\B_{\not\sim A}|"?**
>
> Thank you for noticing this typo. It is fixed in the updated manuscript.
>
> > ### **"we have equality of NP_p(W) = U": shouldn't it be "(U^p-1)^{1/p}"?**
>
> We respectfully disagree. In Equation (6), $U$ already includes the “^{1/p}” term.

---

> > ### Comment · Reviewer_2V1F · 2023-09-25
> >
> > Thank you for your answers to my comments. It is now more clear to me.

---

### Author Response · Authors · 2023-09-21
**Summary of changes.**

> ### **Overall comment**:

We thank the three reviewers for their time and for providing very valuable feedback for improving this paper. We have replied to each comment and question separately below.

> ### **Summary of changes**:
  *  We reduced the number of citations of (Rieck et al., 2019) throughout the paper
  *  We added a reference to (Lacombe et al., 2021) in the Introduction.
  *  We clarified Definition 3.2.
  *  We added a more detailed discussion of “spatial concentration” in Section 3 (page 4).
  *  The terminology for hidden layers and linear transforms (between layers) was unified in Section 4.
  *  We additionally performed two-sample Kolmogorov-Smirnov tests in Section 4 to show that distributions of NP values of permuted and non-permuted weight matrices are different in linear classifiers and in the first layer in deep models, but the tests fail to reject the hypothesis that NP values are identically distributed for later layers.
  *  We provided more details on the relation between our proposed DGP and (Liu et al., 2020) in Section 5.1 and in the Related Work section. In particular, we acknowledge that the filtration underlying DGP is a special case of $\langle \phi=\lambda\rangle$-connected interactions as proposed by Liu et al. (2020).
  *  We included a detailed discussion of the computational complexity of DGP in Section 5.1. Furthermore, we added pseudocode for computing the summary matrix $S$ in Appendix F.
  *  We added a reference regarding the rationale for constructing persistence diagram representation from DGP in Section 5.2.
  *  We provided additional details and a short justification for Remark A.7.
  *  We added figures in Appendix H.2, similar to (Rieck et al., 2019) to show the early stopping results for NP and DGP.

---

### Decision · Action_Editors · 2023-10-17

**Recommendation:** Accept as is

**Comment:**

In this paper the authors consider neural persistence, a network complexity measure rooted in topological data analysis, and show that it is highly correlated with the variance of the network's weights on shallow layers. The authors propose deep graph persistence, a more global version of of neural persistence, which better measures network complexity. No major concerns were raised by the reviewers, and all the minor issues raised by reviewers were satisfactorily addressed by the authors in the rebuttal. I thus recommend acceptance.

**Audience:**

Reviewers also unanimously agree that this paper will be of interest to a subset of TMLR's audience, namely researchers working on topological data analysis and machine learning.

**Claims And Evidence:**

Reviewers unanimously agree that the claims made in this paper are supported by convincing evidence.